# Longitudinal evaluation of fecal microbiota transplantation for ameliorating calf diarrhea and improving growth performance

Hyun Sik Kim[1,5], Tae Woong Whon[1,3,5], Hojun Sung[1], Yun-Seok Jeong[1], Eun Sung Jung [2], Na-Ri Shin[1,4], Dong-Wook Hyun[1], Pil Soo Kim[1], June-Young Lee[1], Choong Hwan Lee[2] & Jin-Woo Bae [1✉]

Calf diarrhea is associated with enteric infections, and also provokes the overuse of antibiotics. Therefore, proper treatment of diarrhea represents a therapeutic challenge in livestock production and public health concerns. Here, we describe the ability of a fecal microbiota transplantation (FMT), to ameliorate diarrhea and restore gut microbial composition in 57 growing calves. We conduct multi-omics analysis of 450 longitudinally collected fecal samples and find that FMT-induced alterations in the gut microbiota (an increase in the family *Porphyromonadaceae*) and metabolomic profile (a reduction in fecal amino acid concentration) strongly correlate with the remission of diarrhea. During the continuous follow-up study over 24 months, we find that FMT improves the growth performance of the cattle. This first FMT trial in ruminants suggest that FMT is capable of ameliorating diarrhea in preweaning calves with alterations in their gut microbiota, and that FMT may have a potential role in the improvement of growth performance.

[1] Department of Life and Nanopharmaceutical Sciences and Department of Biology, Kyung Hee University, Seoul 02447, Republic of Korea. [2] Department of Bioscience and Biotechnology, Konkuk University, Seoul 05029, Republic of Korea. [3] Present address: Microbiology and Functionality Research Group, World Institute of Kimchi, Gwangju 61755, Republic of Korea. [4] Present address: Biological Resource Center, Korea Research Institute of Bioscience and Biotechnology, Jeongeup-Si, Jeollabuk-Do 56212, Republic of Korea. [5] These authors contributed equally: Hyun Sik Kim, Tae Woong Whon. ✉email: baejw@khu.ac.kr

The beef industry has made great improvements in animal facilities, feeding, welfare, breeding, herd management, and the use of biopharmaceuticals. However, the prevention and timely treatment of calf diarrhea still represent a challenge[1]. According to the results of the U.S. National Animal Health Monitoring System study, published in 2018 (Calf Note 203), 39% of calf mortality is caused by diarrhea during the first 3 weeks of life. The causes of calf diarrhea that have been identified are infectious agents, including viruses (for example, the rotavirus group, bovine viral diarrhea virus, and bovine coronavirus), bacteria (for example, *Salmonella* spp. and *Escherichia coli*), and protozoa (for example, *Eimeria zuernii*), and non-infectious factors, such as stress, diet, and humidity[2,3]. Given that most of the financial loss in beef production is attributable to calf morbidity and mortality, a better understanding of the etiology and prevention of diarrhea in preweaning calves is essential to maximize beef production and profitability.

Antibiotics have been widely used to treat or prevent diarrhea and promote growth in livestock[4–6]. Recently, however, accumulating evidence has shown that the use of antibiotics in animal husbandry is associated with many adverse effects. The emergence of antibiotic-resistant bacteria and antibiotic residues in meat are recognized as major problems[7–9]. Due to their potential to kill both pathogenic and beneficial microbes, the use of wide-spectrum antibiotics can promote the colonization of the gut by pathogenic microbes, which can cause disease[10–12]. Thus, antibiotic therapy of diarrheic calves often leads to the recurrence of serious diarrhea within days of starting it. The presence of an imbalance in the intestinal microbial community, followed by the administration of antibiotics, is capable of activating immune responses, inflammation, and peristalsis in the host gut, which causes diarrhea[13,14]. Importantly, a combination of recurrent diarrhea and antibiotic abuse in preweaning calves may result in the immaturity of the ruminal and intestinal microbiota, which has permanent negative effects on the digestion and absorption of dietary components during the fattening period[15–17]. Given that antibiotics also have detrimental impacts on ecology and food safety, their use should be reduced in farming, and alternatives to antibiotic treatment identified.

Several lines of evidence suggest that reconstitution of a healthy microbial community is an effective method of preventing or treating gastrointestinal disorders[18,19]. In particular, fecal microbiota transplantation (FMT, instillation of the feces from a healthy donor into the gastrointestinal tract of a recipient patient) is a highly effective therapy for *Clostridioides difficile* infection (CDI), irritable bowel syndrome, and inflammatory bowel disease (IBD)[20,21]. Microbial controls for the prevention or treatment of gastrointestinal diseases of economic animals as well as human patients have been applied. Recent studies have indicated that the microbiota-derived bacteriocin (i.e., gassericin A) targets the host intestinal epithelium and confers resistance to diarrhea in early weaned piglets, and FMT has the potential to prevent necrotizing enterocolitis in preterm piglets[22,23]. In addition, the transmission of rumen fluid or cud from healthy donor cattle to sick recipient cattle was practiced prior to any understanding of the rumen microflora. Indeed, rumen transformation (the introduction of a wide range of rumen microorganisms, including bacteria, protozoa, fungi, and archaea) has been used to treat animals exposed to botanical toxins[24].

In the present study, we evaluated the therapeutic effect of FMT in calves with diarrhea. We collected fecal samples from healthy donor Korean brown cattle calves (*Bos taurus coreanae*) and transplanted these into recipient calves representing diarrheic progression. We then investigated the temporal changes in the calf gut microbiota in response to FMT by conducting longitudinal metataxonomic and metabolomic analyses of calf fecal samples. Finally, to determine whether the mature gut microbiota instilled by FMT affects the development or growth of calves, their gut microbiota, serum metabolic markers, and body mass gain were followed during the fattening period.

## Results

**FMT ameliorates diarrhea in preweaning calves.** FMT has been applied to monogastric animals, including humans[21,25] and swine[22,23]. However, to our knowledge, no study to date has assessed the use of FMT in ruminants. Prior to assessing the efficacy of FMT for the treatment of calf diarrhea, we performed a preliminary trial to verify the safety of oral FMT (Supplementary Table 1). We collected a fecal sample from a healthy donor and orally administered six randomly selected recipient calves, regardless of the presence of diarrhea, with 0.5 mg/ml fecal concentration twice a day (Supplementary Fig. 1a, b; see "Methods"). We then closely observed these calves for 16 days and found no signs of abnormal behavior or acute illness. Next, we performed 16S rRNA gene amplicon sequencing (hereafter referred to as a metataxonomic analysis) of fecal samples collected per rectum 0, 1, 2, 4, 8, and 16 days post-FMT (Supplementary Fig. 1c) to determine the temporal changes in the calf gut microbiota in response to the FMT treatment. Principal coordinate analysis (PCoA) of the weighted UniFrac distance matrix revealed that the intestinal microbial changes in the recipient's calves were influenced by the microbial composition of the administered feces more than by the feed pellets or maternal milk (Supplementary Fig. 1d, f). Concordant with the shift in the structure of the microbial population in the recipient's calves, the number of amplicon sequence variants (ASVs) shared with the donor's calf was highest 16 days after the start of treatment (Supplementary Fig. 1e). We estimated if microbiota in the gut of the recipient's calves originated from the gut of the donor's calf, or if they were derived from some other external source at during the trial. To demonstrate this capability, we used SourceTracker analysis to quantify the proportions of the different microbial samples (sources) in a target microbial community (sink). SourceTracker analysis showed that the mean contributions of the donor feces to the gut bacterial community of the recipient's calves increased during the 16 days after the initiation of treatment in all the calves and that there was no contribution from external sources other than the donor feces (Supplementary Fig. 2a, b). These results enabled us to expect that oral FMT is capable of affecting the structure of the intestinal microbiota in young ruminants without having negative effects on calf morbidity or mortality.

To further investigate the effects of FMT on the gut microbiota of diarrheic calves, we enlarged both the donor and recipient cohorts of calves. Healthy calves ($n = 6$) that would provide the fecal samples were rigorously selected according to the inclusion and exclusion criteria shown in Table 1. We orally treated the diarrheic recipient calves with 5 g feces as a bolus (0.1 g/ml feces) five times in total (FMT, $n = 20$; Fig. 1a). This total quantity of feces was 12.5 times that used in the preliminary trial. Age-matched diarrheic calves treated with saline (CON, $n = 14$) or antibiotics (ABX, $n = 23$) were included as negative and treatment controls, respectively. We then characterized the severity of diarrhea in the calves on a daily basis and collected fecal samples per rectum 0, 2, 4, 8, 16, 32, and 48 days after the treatments commenced.

Examination of the collected samples showed that FMT, but not saline or antibiotics, reduced the incidence of diarrhea in the recipient calves (Fig. 1b–d). Assessment of the liquidity of the feces, using the Bristol stool scale (BSS), showed that it was significantly lower only on day 48 in the FMT group *vs.* the CON

**Table 1 Inclusion and exclusion criteria for the donor and recipient's calves.**

**Inclusion and exclusion criteria for the recipient's calves**
*Inclusion criteria*
Calves had to meet all the following criteria to be used
1. Male and female calves aged 5–50 days
2. Current moderate-to-severe diarrhea, with a Bristol stool score for the liquidity of 6–7
3. Rectal bleeding and mucosal appearance
*Exclusion criteria*
Calves with any of the following criteria were not used
1. History of antimicrobial agent administration (antibiotics, antifungals, or antivirals)
2. History of communicable livestock diseases
   (e.g., foot-and-mouth disease, contagious bovine pleuropneumonia, anthrax, or brucellosis)
3. Age <5 or >50 days
4. Bristol stool score < 6
**Inclusion and exclusion criteria for healthy fecal donors**
*Inclusion criteria*
Calves had to meet all of the following criteria to be used
1. Male and female calves aged 21–50 days
2. Bristol stool score for the liquidity of 3–4
3. Well, nourished
4. Good physical condition (clean nose, mouth, ears, rump, tail, and hair)
*Exclusion criteria*
Calves with any of the following criteria were not used
1. History of antimicrobial agent administration (antibiotics, antifungals, or antivirals)
2. History of communicable livestock diseases
   (e.g., foot-and-mouth disease, contagious bovine pleuropneumonia, anthrax, or brucellosis)
3. Age <21 or >50 days
4. Current moderate-to-severe diarrhea, with a Bristol stool score for the liquidity of 6–7
5. Rectal bleeding and mucosal appearance
6. Presence of a virus that causes calf diarrhea, determined by the detection of a specific pathogen in the fecal samples
   (group A rotavirus, group B rotavirus, group C rotavirus, bovine coronavirus, bovine torovirus, bovine norovirus, bovine enteric Nebraska-like calicivirus, bovine nebovirus, or bovine viral diarrhea virus)
7. Presence of a bacterium that causes calf diarrhea, determined by the detection of a specific pathogen in the fecal samples
   (*Clostridium perfringens, Salmonella enterica, Salmonella enterica typhimurium*, enterotoxigenic *Escherichia coli*, shigatoxigenic *Escherichia coli*, or enterohemorrhagic *Escherichia coli*)
8. Presence of a protozoan that causes calf diarrhea, determined by the detection of a specific pathogen in the fecal samples
   (*Eimeria zuernii*)

and ABX groups (Fig. 1e). Of the 20 diarrheic calves in the FMT group, 19 calves were in complete remission 48 days after the FMT treatment (95.0%), and none of the calves had died (mortality rate, 0%). However, in the CON and ABX groups, only 5 (35.7%) and 6 (26.1%) calves were in complete remission, and 2 (mortality rate, 14.3%) and 4 (mortality rate, 17.4%) had died 48 days after the saline and ABX treatments, respectively (Table 2 and Supplementary Tables 2–4). Collectively, these data suggest that FMT reduces both the incidence of diarrhea and diarrhea-related mortality in calves, without causing adverse events.

**FMT alters the intestinal microbiota of diarrheic calves**. We next assessed the gut microbial profile of the preweaning calves following FMT. We collected fecal samples per rectum from the FMT ($n = 20$; 147 samples), CON ($n = 14$; 96 samples), and ABX ($n = 23$; 157 samples) calves at a series of time points and performed a metataxonomic analysis. PCoA of the weighted UniFrac distance matrix revealed widely dispersed data points on plots of the diarrheic samples on day 0, regardless of the treatment group, implying that there was severe microbial dysbiosis in the gut of the diarrheic calves (Fig. 2a–c). Interestingly, the structure of the microbial community in the FMT calves progressively changed to resemble that of the donor's calves during the experimental period (Fig. 2a, d), whereas the CON and ABX calves exhibited neither a directional change of their microbial community structure nor the presence of mature gut microbiota as the calf

aged, as evidenced by the widely scattered data points on the PCoA plots of the end point fecal samples (day 48) (Fig. 2b, c). These results suggest that the amelioration of diarrhea observed only in the FMT calves (as shown in Fig. 1) might be a consequence of the gut microbial eubiosis induced by FMT.

Analysis of the UniFrac distance within the groups revealed that the microbial dissimilarity values for the FMT calves, but not the calves in the CON and ABX groups, gradually decreased to the levels of the healthy donors (Fig. 2d–f). When the dissimilarity values of the day 48 samples were intra-compared, the FMT group had the lowest value of the three, which suggests that the structure of the microbial community in the diarrheic calves stabilized following the FMT (Fig. 2g). We next compared the UniFrac distance of the healthy donor samples with those of the FMT, CON, and ABX calves, and found relatively high dissimilarity values for the fecal microbiota of the donors and FMT calves at day 0, but this significantly decreased with time (Fig. 2h). However, the dissimilarity values of the donors and the CON or ABX calves did not change or tended to increase with time (Fig. 2i, j). Forty-eight days after the treatments commenced, the FMT calves had the lowest dissimilarity values (Fig. 2k) and the highest number of ASVs shared with the healthy donors, which demonstrates that the microbial community of the healthy donors was effectively transmitted by FMT (Supplementary Fig. 3a, b).

SourceTracker analysis was used to determine whether the gut microbiota of CON, ABX, and FMT calves originated from the

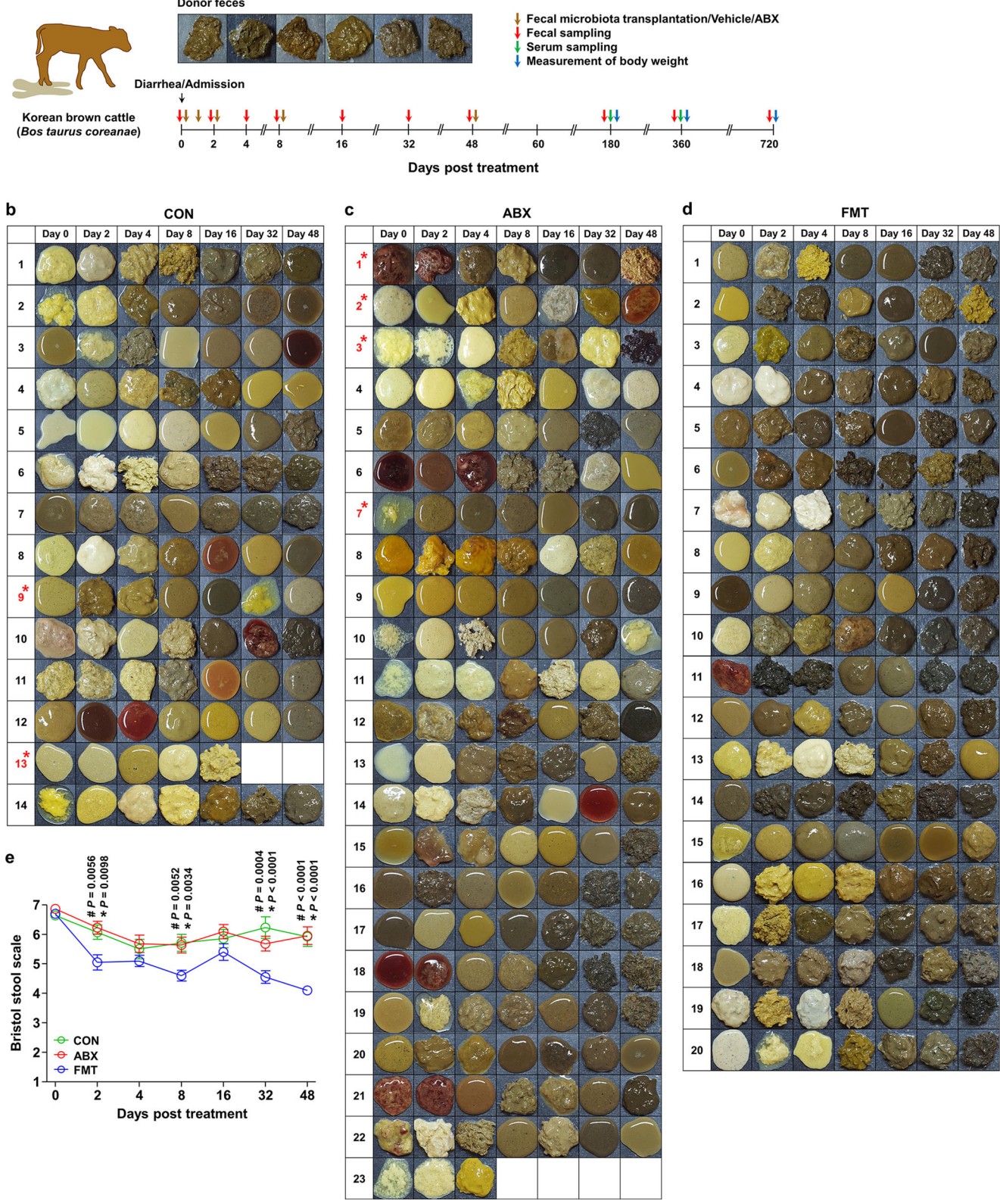

**Fig. 1 Design of the FMT trials and the effect of the treatment on the incidence of diarrhea. a** Design of the FMT trial in calves with diarrhea. A detailed description is provided in the Methods section. **b–d** Images of the fecal samples collected from the recta of CON (**b**, $n = 14$), ABX (**c**, $n = 23$), and FMT (**d**, $n = 20$) calves 0, 2, 4, 8, 16, 32, and 48 days after the treatments commenced. A red asterisk indicates a calf that died. **e** The incidence of diarrhea is represented by the BSS of the fecal samples collected from CON ($n = 14$), ABX ($n = 23$), and FMT ($n = 20$) calves. The BSSs for the CON, ABX, and FMT groups are shown in green, red, and blue, respectively. Data are shown as mean ± SEM. The $P$ values were determined using the Mann–Whitney $U$ test (two-tailed). *Comparison of the CON and FMT groups; #comparison of the ABX and FMT groups; and †comparison of the CON and ABX groups. CON control, ABX antibiotic, FMT fecal microbiota transplantation, BSS Bristol stool scale. Source data are provided as a Source Data file.

**Table 2 The mortality rate (%) and the complete remission rate (%) of the diarrheic calves, according to treatment.**

|  | % Percentage (n) |
|---|---|
| CON |  |
| Mortality rate | 14.3% (2/14) |
| Complete remission rate | 35.7% (5/14) |
| ABX |  |
| Mortality rate | 17.4% (4/23) |
| Complete remission rate | 26.1% (6/23) |
| FMT |  |
| Mortality rate | 0% (0/20) |
| Complete remission rate | 95.0% (19/20) |

CON control, ABX antibiotic, FMT fecal microbiota transplantation.

gut of a donor's calves during the trial period. The results show that the mean contributions of the donor feces to the gut bacterial communities of CON, ABX, and FMT calves were similar on day 0 (Fig. 2l and Supplementary Fig. 3c). Thus, these intestinal ASVs might represent the core microbiota that is inherited from the mother, and this core microbiota has frequently been identified during the development of newborn calves. However, the mean contributions of the donor feces to the gut bacterial community of the CON, ABX, and FMT calves were highest 48 days after the initiation of treatment in FMT calves (Fig. 2l and Supplementary Fig. 3d). Thus, FMT reduced interindividual variation in the microbial community and induced a change in the microbial structure of diarrheic calves toward that of the healthy donor calves.

**FMT results in gradual maturation of the gut microbiota in diarrheic calves.** We next assessed the abundance of microbial taxa in the guts of CON, ABX, and FMT calves at day 48. We hypothesized that FMT-induced remission of diarrhea might be the result of the generation of a eubiotic state, which is characterized by resilience and resistance to external and endogenous disturbances[26,27]. Linear discriminant analysis of the day 48 metataxonomic data, coupled with effect size measurements (LEfSe), was used to generate a circular cladogram, which indicates that the phyla Verrucomicrobia, Proteobacteria, and Bacteroidetes were the discriminatory taxa of the CON, ABX, and FMT calves, respectively (Fig. 3a). The relative abundances of the bacterial taxa given by the LEfSe were then compared. In the ASV feature table that was compiled using the SILVA database (version 132), it was apparent that sequences assigned to the family *Verrucomicrobiaceae* predominated in the CON calves (Fig. 3b and Supplementary Fig. 4a), whereas sequences assigned to the families *Enterobacteriaceae* and *Lactobacillaceae* were significantly enriched in the ABX calves (Fig. 3c and Supplementary Fig. 4b), and FMT was associated with significantly larger populations of the *Ruminococcaceae*, *Bacteroidaceae*, *Paraprevotellaceae*, and *Porphyromonadaceae* (genus *Parabacteroides*) and families (Fig. 3d and Supplementary Fig. 4c).

Facultative anaerobic Gram-negative bacteria (e.g., Proteobacteria) are usually present in small numbers in the gut of healthy adult subjects, where strictly anaerobic bacteria (e.g., Bacteroidetes and Firmicutes) tend to predominate[28]. However, in mammalian neonates, facultative anaerobes derived from maternal sources consume oxygen in the gut during the genesis of the initial gut microbiota, which encourages the settlement of strict anaerobes[29–31]. To determine whether normal gut microbial maturation is induced by FMT in diarrheic calves, we assessed the temporal changes in the major microbial taxa belonging to the

phyla Firmicutes, Bacteroidetes, Proteobacteria, Verrucomicrobia, and Actinobacteria (Supplementary Fig. 5a–e), and belonging to the families *Ruminococcaceae*, *Bacteroidaceae*, *Enterobacteriaceae*, *Lachnospiraceae*, *Lactobacillaceae*, *Porphyromonadaceae*, and *Paraprevotellaceae* (Fig. 3e and Supplementary Fig. 5f–j). Our data showed that the relative abundances of the families *Enterobacteriaceae* and *Porphyromonadaceae* in the diarrheic samples on day 0 did not differ between the groups (Fig. 3e). However, the abundance of the *Enterobacteriaceae* gradually decreased with time, and that of the *Porphyromonadaceae* gradually increased. We also found that there was a significantly lower relative abundance of *Enterobacteriaceae*, and a significantly higher relative abundance of *Porphyromonadaceae*, in the FMT group than in the CON and ABX groups on days 32 and 48. At the family level, there was no noticeable change in the abundance other than the above two taxa.

We next conducted multiple variable correlation analysis between the BSS and the relative abundances of the families *Porphyromonadaceae* or *Enterobacteriaceae*. In calves of the CON, ABX, and FMT groups, strong positive correlations were identified between the relative abundance of the *Enterobacteriaceae* and the BSS for all the groups (CON, Pearson $r = 0.673$, $P = 0.049$; ABX, Pearson $r = 0.932$, $P = 0.001$; FMT, Pearson $r = 0.834$, $P = 0.010$; Fig. 3f). The above-described correlation analysis suggests that gut dysbiosis exemplified by an abnormal increase in the abundance of *Enterobacteriaceae* is highly likely to trigger diarrhea in young calves. However, the relative abundance of the *Porphyromonadaceae* and BSS were found to be significantly negatively correlated only in the FMT group (CON, Pearson $r = -0.203$, $P = 0.337$; ABX, Pearson $r = -0.372$, $P = 0.256$; FMT, Pearson $r = -0.714$, $P = 0.041$). In addition, a negative correlation was identified between the relative abundance of the *Enterobacteriaceae* and that of the *Porphyromonadaceae* in the FMT group alone (CON, Pearson $r = -0.752$, $P = 0.025$; ABX, Pearson $r = -0.491$, $P = 0.130$; FMT, Pearson $r = -0.824$, $P = 0.024$). These results indicate the maturation of the gut microbiota following FMT, characterized by an increase in the abundance of the *Porphyromonadaceae*, which restricts the expansion of the *Enterobacteriaceae* population and is associated with diarrheic remission.

**FMT alters the fecal metabolomic profile of the diarrheic calves.** For the future application of FMT in the field, it is particularly important to develop an understanding of how microorganisms and microbial products affect the incidence of calf diarrhea[32]. To evaluate the effects of changes in the gut microbiota on the intestinal microenvironment, we next analyzed the fecal metabolomes of calves before (day 0, $n = 54$) and after (day 48, $n = 54$) treatment using gas chromatography time-of-flight mass spectrometry (GC–TOF–MS). Consistent with the intestinal microbial metataxonomic profiles, PCoA of the Bray–Curtis dissimilarity matrix revealed widely dispersed data points on plots of the fecal metabolomes from pre-treatment diarrheic calves, regardless of their allocated group (Fig. 4a). Interestingly, however, we observed that the data points corresponding to the FMT group closely clustered plots among the post-treatment samples. The points corresponding to FMT samples significantly separated from those corresponding to the CON and ABX groups (permutational multivariate analysis of variance [PERMANOVA]: $P = 0.007$ for a comparison of the CON and FMT calves and $P = 0.001$ for a comparison of the ABX and FMT calves; Fig. 4b).

The fecal metabolites in the post-treatment calves were clustered using a Bray–Curtis dissimilarity matrix-based UPGMA dendrogram, and the relative quantities of metabolites were

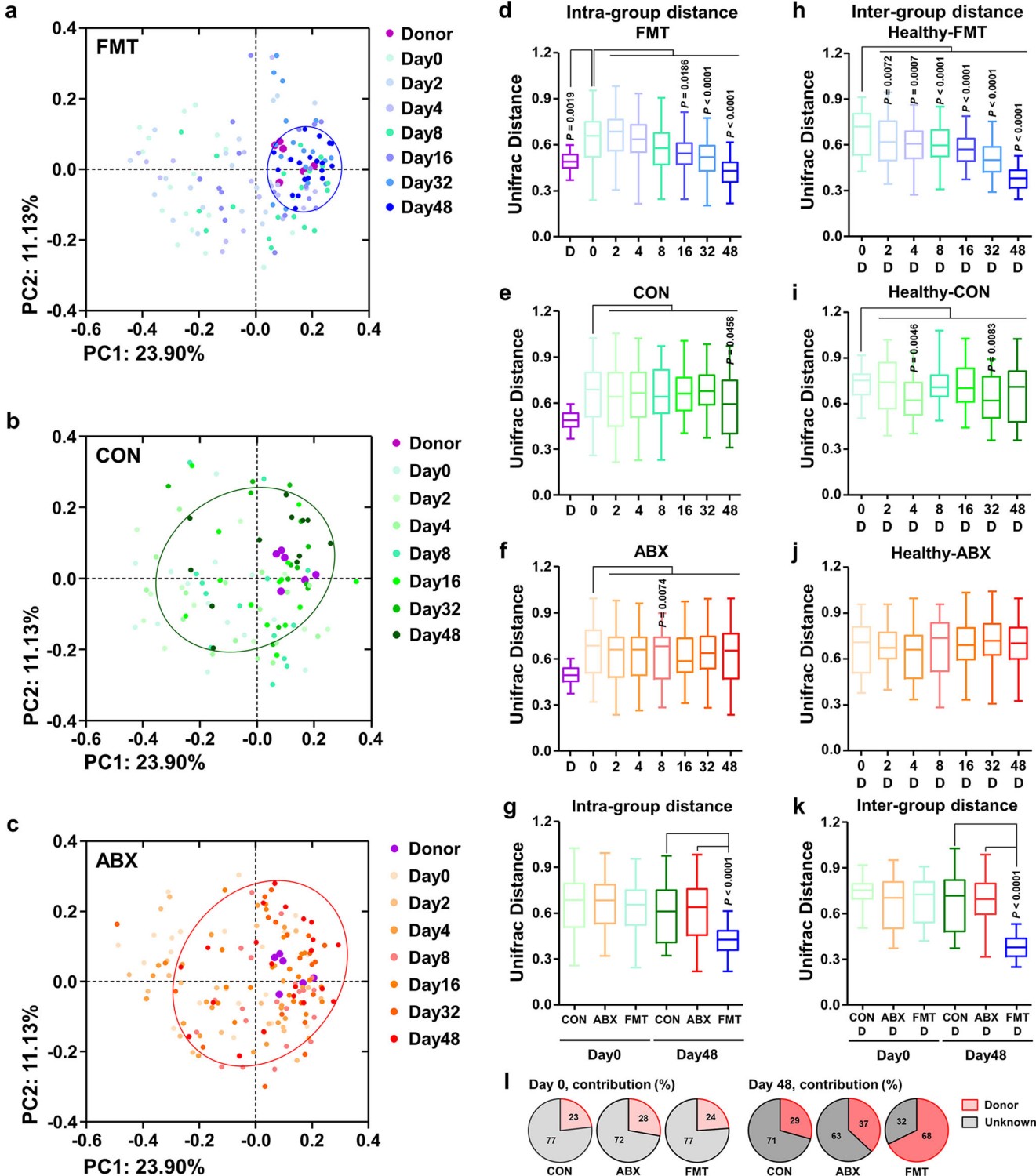

visualized using a heatmap (Fig. 4c). The concentrations of several amino acids (alanine, leucine, valine, isoleucine, glycine, arginine, ornithine, and glutamic acid) were relatively low in the day 48 FMT calves when compared with those in the day 48 CON and ABX calves (Supplementary Fig. 6). Indeed, these relative amounts had significantly decreased from baseline only in FMT calves before the 48 day time point (Fig. 4d). By contrast, there were relatively higher concentrations of several amino acids (alanine, valine, isoleucine, and glycine) 48 days after treatment commenced in the CON and ABX groups. Therefore, we

quantified fecal amino acid and branched-chain amino acid (BCAA; isoleucine, leucine, and valine) concentrations using an enzyme-based assay, and found that the concentrations of both amino acids (Fig. 4e) and BCAAs (Fig. 4f) were significantly lower in the FMT calves than in the CON and ABX calves at day 48. Taken together, these results suggest that the remission of diarrhea in calves that undergo FMT is accompanied by changes in their metabolomes, and in particular by decreases in amino acid concentrations, which are concomitant with changes in the gut microbiome.

**Fig. 2 Changes in the fecal bacterial profiles of diarrheic calves following the commencement of treatment. a–c** PCoA, based on the weighted UniFrac distance matrix, of the bacterial 16S rRNA gene sequence data for fecal samples collected 0, 2, 4, 8, 16, 32, and 48 days after the start of treatment are shown for CON ($n = 108$), ABX ($n = 176$), and FMT ($n = 166$) calves. The PCoA scatterplot shows the bacterial communities in the CON, ABX, and FMT calves in blue, red, and green, respectively. **d–f** Within-group microbial dissimilarity values (calculated based on the weighted UniFrac distance matrix) for the FMT (**d**, $n = 20$), CON (**e**, $n = 14$), and ABX (**f**, $n = 23$) calves 0, 2, 4, 8, 16, 32, and 48 days after the start of treatment. **g** Within-group microbial dissimilarity values for the FMT ($n = 20$), CON ($n = 14$), and ABX ($n = 23$) calves on days 0 and 48. **h–j** Intergroup microbial dissimilarity values (calculated based on the weighted UniFrac distance matrix) for healthy donor calves and FMT (**h**, $n = 20$), CON (**I**, $n = 14$), and ABX (**j**, $n = 23$) calves 0, 2, 4, 8, 16, 32, and 48 days after the start of treatment. **k** Intergroup microbial dissimilarity values for the healthy donor calves and the CON ($n = 14$), ABX ($n = 23$), and FMT ($n = 20$) calves on days 0 and 48. Microbial dissimilarity values, based on weighted UniFrac distance matrices, for each group, are presented as box plots. The lines, boxes, and whiskers in the box plots represent the median, and 25th, and 75th percentiles, and the min-to-max distribution of replicate values, respectively. **l** Results of Sourcetracker analysis showing the average contributions of donor feces to the bacterial communities in the CON ($n = 14$), ABX ($n = 23$), and FMT ($n = 20$) calves on days 0 and 48. Source: gut microbiome of donor's calves; sink: CON, ABX, and FMT calves. The $P$ values were determined using the Mann–Whitney $U$ test (two-tailed) (**d–l**). CON control, ABX antibiotic, FMT fecal microbiota transplantation, D donor. Source data are provided as a Source Data file.

**The FMT-mediated gut microbial assembly further affects host growth performance.** In ruminants, production efficiency is linked to the composition of the ruminal microbiome, as previously illustrated by the identification of an association between microbial components and residual feed intake[33,34]. Therefore, the characterization and understanding of the role of environmental factors (diets, the microbiome, and treatments) that can affect the growth of production animals, including ruminants, is important both scientifically and economically. To determine whether calf growth is affected by FMT, we next measured the body mass of the treated calves at 6, 12, and 24 months of age. The 6-month-old cattle (both male and female) showed no differences in body mass between the groups (Fig. 5a), but both male and female 12-month-old FMT-treated cattle had significantly higher body masses than gender-matched CON and ABX cattle (Fig. 5b). Moreover, the 24-month-old FMT-treated cattle had significantly higher body masses (Fig. 5c) and dressed masses (Fig. 5d) than gender-matched CON and ABX cattle. We then calculated the body mass gain of the cattle during their 6 months (from 6 to 12 month old) and 12 months (from 12 to 24 months old), and found that both male and female cattle in the FMT group had gained significantly more body mass than gender-matched ABX cattle (Supplementary Fig. 7a, b). In addition, there was a significant difference in body mass gain between the male FMT and gender-matched CON cattle (Supplementary Fig. 7a, b).

We created metataxonomic profiles for the gut microbiota in the 12-month-old cattle ($n = 51$) in which the overall body mass was increased by FMT. PCoA of the weighted UniFrac distance matrix showed that the FMT samples clustered separately from the CON and ABX samples, which clustered together (PERMANOVA: $P = 0.001$ for the comparison of the CON and FMT cattle, $P = 0.001$ for the comparison of the ABX and FMT cattle, and $P = 0.375$ for the comparison of the CON and ABX cattle; Fig. 5e). LEfSe was used to identify the bacterial families that were enriched in each group of cattle. This showed that the *Bacteroidaceae* and *Porphyromonadaceae* were more abundant in the CON cattle, the *Lachnospiraceae* were more abundant in the ABX cattle, and the *Christensenellaceae*, *Clostridiaceae*, *Peptostreptococcaceae*, *Dehalobacteriaceae*, and *Coriobacteriaceae* were more abundant in the FMT cattle (Fig. 5f). The relative abundances of the bacterial taxa by the LEfSe were then compared (Fig. 5g–i) and indicated that sequences assigned to the families *Christensenellaceae*, *Clostridiaceae*, *Peptostreptococcaceae*, *Dehalobacteriaceae*, and *Coriobacteriaceae* predominated in the FMT cattle (Fig. 5i and Supplementary Fig. 8a), which suggests that microbes abundant in the gut of FMT cattle may be associated with growth and/or fattening in growing cattle.

Subsequently, a PICRUSt (Phylogenetic Investigation of Communities by Reconstruction of Unobserved States) pipeline[35] was used to determine whether different microbial groups in the three groups might be associated with differences in functional profiles. The gene families predicted using PICRUSt revealed enrichment of features required for fast growth (e.g., ribosome, DNA repair and recombination proteins, and DNA replication proteins) and BCAA biosynthesis (valine, leucine, and isoleucine biosynthesis) in the FMT cattle (Supplementary Fig. 8b). In summary, various microbial groups formed through FMT may be able to respond more flexibly to dietary intervention during fattening.

**The long-term consequences of FMT for the systemic metabolome in fattening cattle.** As shown in Fig. 4, FMT-induced changes in the gut microbiome also affected the fecal metabolomes of the calves locally. To interrogate whether FMT-induced changes in the gut microbial assembly persistently affect systemic metabolism in growing cattle, we analyzed the serum metabolome of 12-month-old cattle ($n = 50$) using GC–TOF–MS. In accordance with the intestinal metabolome profiles of the day 48 samples, PCoA of the Bray–Curtis dissimilarity matrix showed segregation of serum metabolites according to treatment (PERMANOVA: $P = 0.001$ for the comparison of the CON and FMT cattle, $P = 0.031$ for the comparison of the CON and ABX cattle, and $P = 0.001$ for the comparison of the ABX and FMT cattle; Fig. 6a). The UPGMA dendrogram and heatmap analysis showed that the FMT group had a distinct serum metabolic profile from the other groups (Fig. 6b). There were significantly higher relative concentrations of serum BCAAs in the FMT cattle than in the CON or ABX cattle ($P < 0.01$; Supplementary Fig. 9). To corroborate the metabolomic data, enzyme assays were performed to determine the concentrations of BCAAs in the sera of 6-, 12-, and 24-month-old cattle. Consistent with the results of the metabolomic analysis, the concentration of BCAAs was significantly higher in the serum of FMT cattle than in the other groups, regardless of their age (Fig. 6c). These data demonstrate that FMT has significant effects on the systemic metabolome of cattle that might contribute to the observed difference in the growth of the cattle.

In summary, in this well-controlled intervention study conducted in preweaning calves with diarrhea, we have shown that a healthy gut microbiome in donor's calves is an important mediator of the effect of FMT to induce the remission of calf diarrhea, and it led to the production of a distinct set of microbial metabolites. The findings also suggest that FMT-induced changes in the gut microbiota exert improvement of host growth performance independent of developmental age.

## Discussion

"Eubiotic status" of the gut microbiota is characterized by a preponderance of several species that belong mainly to two

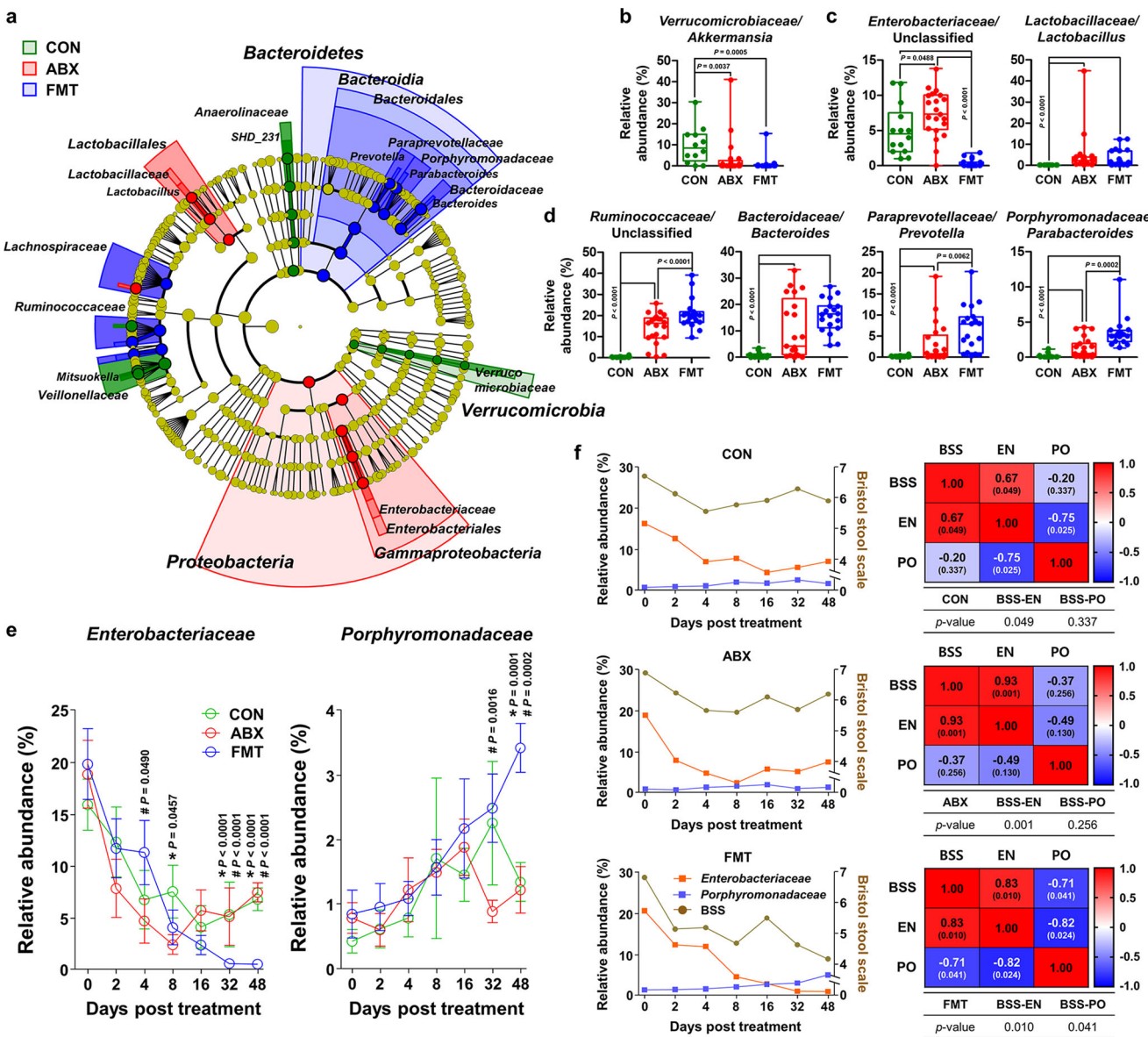

**Fig. 3 Discriminating microbial taxa associated with the incidence of diarrhea, and correlation analysis of gut *Enterobacteriaceae* or *Porphyromonadaceae* with the incidence of diarrhea. a** The linear discriminant analysis effect size (LEfSe) circular cladogram generated from the day 48 metataxonomic data identified the discriminating taxa for the CON (*n* = 14), ABX (*n* = 23), and FMT (*n* = 20) groups (shown in different colors). **b–d** Relative abundance of the predominant taxa in ASV feature table compiled using the SILVA database for the CON (**b**, *n* = 14), ABX (**c**, *n* = 23), and FMT (**d**, *n* = 20) groups, presented as bar charts and dot plots. The lines, boxes, and whiskers in the box plots represent the median, and 25th, and 75th percentiles, and the min-to-max distribution of replicate values, respectively. Data are shown as mean ± SEM and were analyzed using the Mann–Whitney *U* test (two-tailed). **e** Changes in the relative abundance of the family *Enterobacteriaceae* (left) and *Porphyromonadaceae* (right) with time (CON, *n* = 14; ABX, *n* = 23; FMT, *n* = 20). Data are shown as mean ± SEM. The *P* values were determined using the Mann–Whitney *U* test (two-tailed). *Comparison of the CON and FMT groups; #comparison of the ABX and FMT groups; and †comparison of the CON and ABX groups. **f** The relative abundances of the families *Enterobacteriaceae* (left *y*-axis, orange) and *Porphyromonadaceae* (left *y*-axis, blue), and the BSS (right *y*-axis, brown) in each group over time (left panel). The Pearson correlation coefficients for multiple variants and statistical significance (two-tailed) for the relationships among BSS and the relative abundances of the families *Enterobacteriaceae* and *Porphyromonadaceae* were calculated (right panel). CON control, ABX antibiotic, FMT fecal microbiota transplantation, BSS Bristol stool scale, EN *Enterobacteriaceae*, PO *Porphyromonadaceae*. Source data are provided as a Source Data file.

bacterial phyla: the Firmicutes and Bacteroides, while potentially pathogenic species, such as those belonging to the phylum Proteobacteria (family *Enterobacteriaceae*) are present in very low numbers[26,27]. Ali Metchnikoff has suggested that most disease begins in the digestive tract when the "good" bacteria are no longer able to control the "bad" ones[36]. He conceptualized this situation as dysbiosis, which implies a disruption of the mutually harmonious gut ecosystem. Given that it is very difficult to eliminate specific harmful microorganisms using antibiotics,

many scientists and clinicians agree that alternative therapies are required. Unfortunately, although attempts have been made to treat gastrointestinal diseases by administering livestock a fecal inoculum or specific bacterial species, no previous studies have shown meaningful changes in the gut microbial community[22,23]. Here, we have shown that FMT reduces the incidence of diarrhea and changes gut environmental conditions from a dysbiotic status to a eubiotic status in pre-weaning diarrheic calves. Indeed, the intestinal microbiome of the recipient's calves gradually became

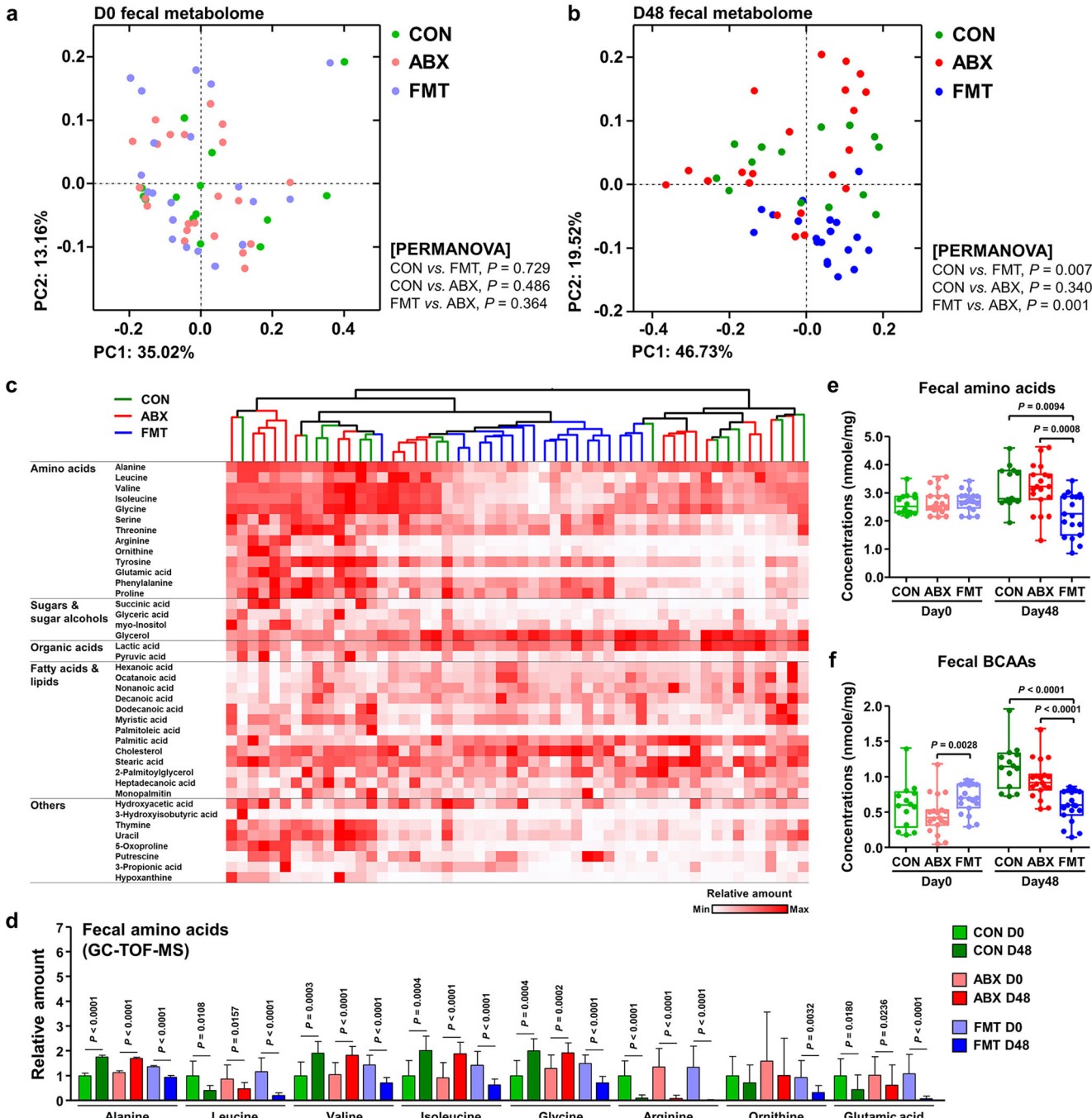

**Fig. 4 Changes in the fecal metabolome profile of the diarrheic calves following FMT.** The metabolomes of the calves were analyzed using GC-TOF-MS and clustered using PCoA, based on the Bray–Curtis dissimilarity matrix. **a** The fecal metabolome profiles of calves before (day 0, $n = 54$) and **b** after (day 48, $n = 54$) treatment. The metabolome profiles for the CON, ABX, and FMT groups are shown in the same colors. The data were analyzed using PERMANOVA, with 999 permutations. **c** The abundant metabolites in day 48 samples were clustered using a UPGMA dendrogram, based on the Bray–Curtis dissimilarity matrix, and the relative abundances are represented in a heatmap. Samples from the same groups are shown in the same color. **d** The relative quantities of rectal metabolites are displayed as bar charts and dot plots (day 0, $n = 54$; day 48, $n = 54$). **e** Amino acid concentrations in the fecal samples obtained from calves in each group on days 0 ($n = 54$) and 48 ($n = 54$). The amino acid concentrations were measured by an enzymatic method using an L-amino acid quantitation kit, and are displayed as box and dot plots. The lines, boxes, and whiskers in the box plots represent the median, and 25th, and 75th percentiles, and the min-to-max distribution of replicate values, respectively. **f** The branched-chain amino acid (BCAA) concentrations in the fecal samples obtained from calves at days 0 ($n = 54$) and 48 ($n = 54$). The concentrations of the BCAAs are displayed as box and dot plots. The lines, boxes, and whiskers in the box plots represent the median, and 25th, and 75th percentiles, and the min-to-max distribution of replicate values, respectively. Data are shown as mean ± SEM and were analyzed using the Mann–Whitney $U$ test (two-tailed). CON control, ABX antibiotic, FMT fecal microbiota transplantation, BCAAs branched-chain amino acids. Source data are provided as a Source Data file.

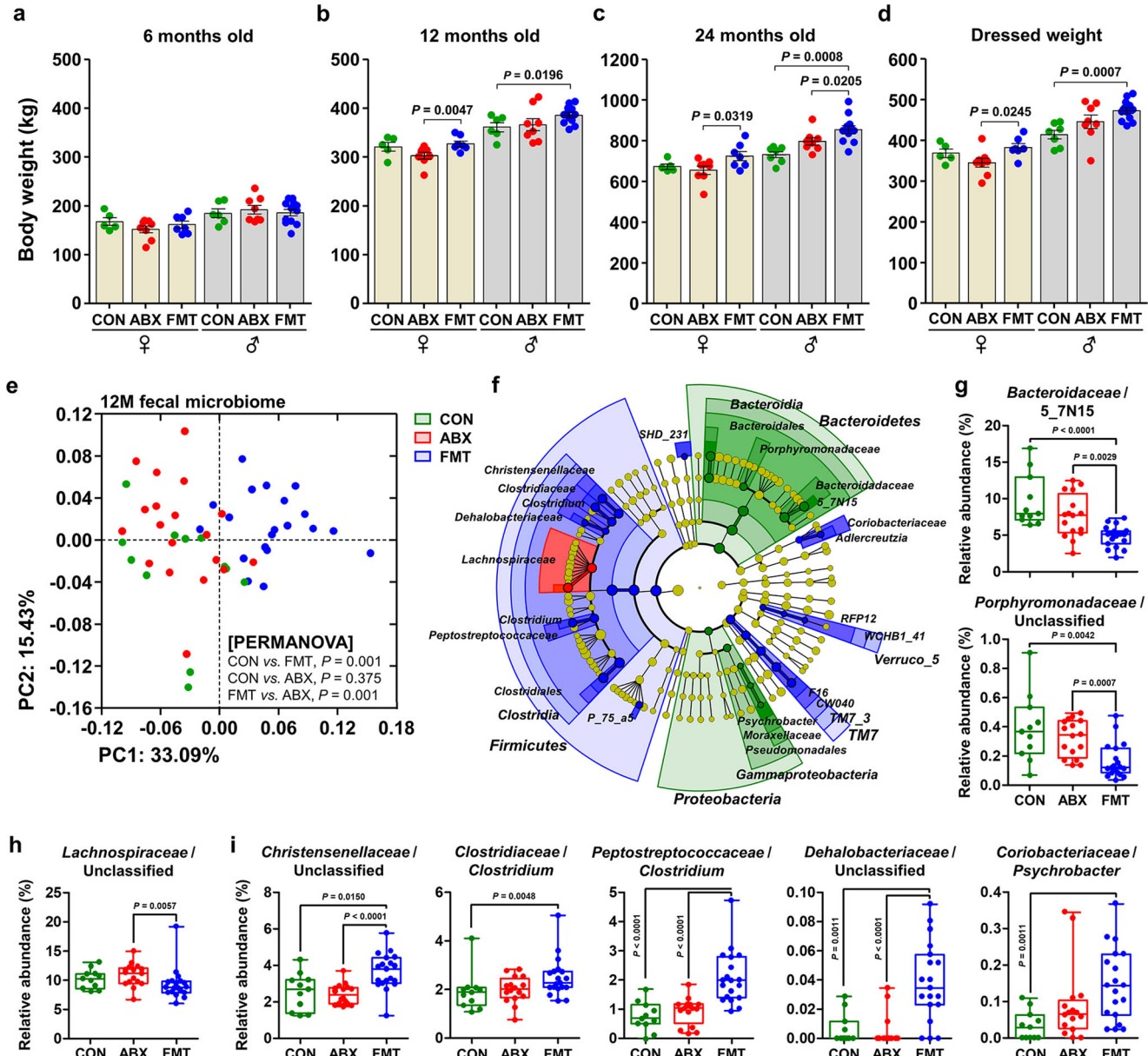

**Fig. 5 The effect of FMT-driven gut microbial assembly on growth performance. a–c** Body mass of the cattle ($n = 50$) at 6 (**a**), 12 (**b**), and 24 (**c**) months of age. **d** Dressed mass of the cattle at 24 months of age ($n = 50$). Bar graphs and dot plots are colored according to sex and treatment. **e** Metataxonomic profiles of the gut microbiome in the 12-month-old cattle ($n = 50$). PCoA of the rectal bacterial 16S rRNA gene sequences, based on the weighted UniFrac distance matrix. The data were analyzed using PERMANOVA, with 999 permutations. **f** The enriched microbial taxa in each group of 12-month-old cattle were identified using LEfSe and are presented using the circular cladogram. The discriminating taxa for each group are denoted in different colors. **g–i** Relative abundances of the discriminating taxa were compiled in the feature ASV table, based on the SILVA database for the CON (**g**, $n = 11$), ABX (**h**, $n = 19$), and FMT (**i**, $n = 20$) groups, and are presented as box and dot plots. The lines, boxes, and whiskers in the box plots represent the median, and 25th, and 75th percentiles, and the min-to-max distribution of replicate values, respectively. Data are shown as mean ± SEM and were analyzed using the Mann–Whitney $U$ test (two-tailed). FMT fecal microbiota transplantation. Source data are provided as a Source Data file.

similar to that of the donor's calves following FMT, and there was a particular increase in the relative abundance of the *Porphyromonadaceae* (Figs. 2 and 3).

Many species belonging to the family *Porphyromonadaceae* are part of the indigenous microbiota of the human and animal gastrointestinal tracts and oral cavities, but some species in this family are commonly associated with a variety of human and animal infections[37]. In mice, members of this family have anti-inflammatory effects and protect the gut against both bacterial infection and colorectal cancer[38,39]. However, their roles in the human gut are more of a mixed bag. The abundance of the family

*Porphyromonadaceae* decreases in the guts of overweight individuals that are pursuing effective weight-loss treatment[40]. Moreover, when the 16S rRNA genes in the feces of 338 people (including cases with CDI, diarrheic controls, and non-diarrheic controls) were analyzed, the family *Porphyromonadaceae* was largely absent in CDI cases and highly associated with non-diarrheic controls[41]. However, when inflammation in the liver of cirrhosis patients prevents bile acids from entering the small intestine, these microbes can overgrow, release more pro-inflammatory signals, and create a positive feedback loop that causes further liver damage[42,43]. In the present study, when

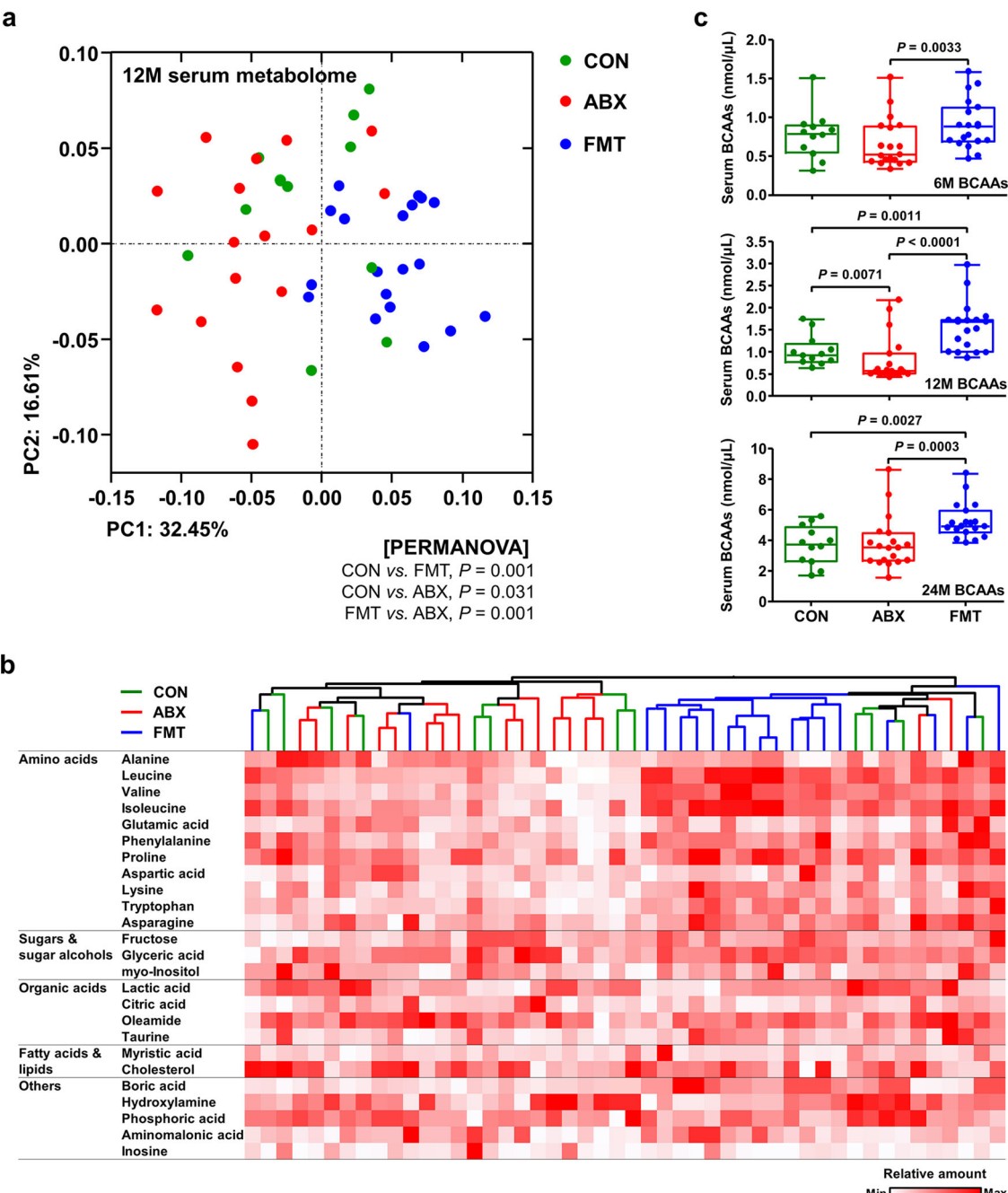

**Fig. 6 Serum metabolome profiles of the 12-month-old cattle. a** The serum metabolomes of the 12-month-old cattle ($n = 50$) were analyzed using GC–TOF–MS and clustered using PCoA, based on the Bray–Curtis dissimilarity matrix. The data were analyzed using PERMANOVA, with 999 permutations. **b** The abundant serum metabolites in samples from 12-month-old cattle were clustered using the UPGMA dendrogram, based on the Bray–Curtis dissimilarity matrix, and the relative quantities are represented in a heatmap. Samples from the same groups are the same color. **c** Serum branched-chain amino acid (BCAA) concentrations in the 6- (upper, $n = 50$), 12- (middle, $n = 50$), and 24- (lower, $n = 50$) month-old cattle, quantified enzymatically and displayed as box and dot plots, as described in the "Methods" section. The lines, boxes, and whiskers in the box plots represent the median, and 25th, and 75th percentiles, and the min-to-max distribution of replicate values, respectively. Data are shown as mean ± SEM and were analyzed using the Mann–Whitney $U$ test (two-tailed). BCAAs, branched-chain amino acids. Source data are provided as a Source Data file.

compared with the diarrheic calves, the gut microbiota of the FMT calves at day 48 included larger numbers of the family *Porphyromonadaceae*, the abundance of which negatively correlated with the incidence of diarrhea. These findings collectively suggest that regulating the quantitative changes of the family *Porphyromonadaceae* in the intestines of young calves may be a cornerstone for resolving calf diarrhea.

We found that the fecal concentrations of amino acids (alanine, leucine, valine, isoleucine, glycine, arginine, ornithine, and glutamic acid) decrease as a result of FMT (Fig. 4), which suggests that the metabolism of microbes associated with the remission of diarrhea leads to low fecal concentrations of amino acids. Amino acids are necessary for intestinal growth, the maintenance of mucosal integrity, and barrier function. They are essential

precursors for glutathione, polyamines, nitric oxide, and other molecules in intestinal epithelial cells, and are building blocks for the macromolecular synthesis that is necessary for mucosal wound healing and energy production in intestinal cells[44]. Gut bacteria can use amino acids to synthesize proteins and other metabolites, which play important roles in the nutrition and physiology of the host[45–47]. Studies conducted in mice have shown that gut bacteria alter the distribution of free amino acids in the gastrointestinal tract and affect the bioavailability of amino acids for the host[48]. When the gut microbiota is healthy (eubiotic status), a trophic network is maintained that is suitable for the fermentation of substances present in the intestine, but when the gut microbial balance is disrupted (dysbiotic status), this trophic network is disrupted and the community shows poor fermentation efficiency[49]. Thus, higher concentrations of amino acids in the intestine are thought to be the result of incomplete fermentation. Similarly, the results of pediatric studies have shown that fecal free amino acid concentrations are tenfold higher during diarrhea than those during periods of remission or in normal individuals[50,51]. Furthermore, in a recent study, the dysbiotic gut microbiota of patients with diarrhea was found to be characterized by higher concentrations of free amino acids, especially of proline, and was associated with higher susceptibility to CDI[52]. In addition, previous studies conducted in rodents and pigs have provided evidence that certain amino acids, particularly glutamine and arginine, may influence the progression of IBD[53] and reduce inflammation and oxidative stress. Amino acid supplementation may be beneficial for patients with IBD or cancer, but it may also have adverse effects in the human bowel. Administration of amino acids such as arginine, cysteine, ornithine, and citrulline causes a variety of gastrointestinal side effects, including nausea, diarrhea, abdominal cramping, and bloating[54]. Furthermore, an amino acid imbalance or an increase in protein catabolism can provoke metabolic acidosis, which is also termed hyperchloremic acidosis, and the most common underlying mechanism involves the loss of large amounts of the base because of diarrhea[55,56]. In light of these previous findings, it is likely that microbial dysbiosis, which leads to an imbalance in the intestinal amino acids, is associated with a higher risk of diarrhea in calves. Further studies should aim to determine in more detail the effects of amino acid metabolism on gastrointestinal disorders in livestock.

The present study is the first to use FMT to treat calf diarrhea in place of antibiotics. However, for FMT to be used clinically as a treatment of calf diarrhea, it is necessary to more fully characterize its effects in the gut[57]. First, the underlying microbiological mechanisms whereby members of the family *Porphyromonadaceae* cause the remission of calf diarrhea must be more fully understood. Second, the accurate taxonomy of the specific members of the *Porphyromonadaceae* that are involved should be characterized and they should be isolated and cultured, such that their functional roles in the host can be determined. Third, the effectiveness of FMT should be validated in larger numbers of animals that exhibit moderate-to-severe diarrheic symptoms, and the host baseline physiology, characteristics, and gut microbiota should be characterized in more detail. In particular, the specific bacterial species associated with a positive or negative response should be identified and isolated, which would permit customized FMT or defined microbial manipulations in the future. As an illustration of the importance of this, in human IBD and ulcerative colitis patient cohort studies in which FMT was performed equally, the effects depended on the gut microbial structure and/or the physiological characteristics of recipients[58,59]. Encouragingly, gut colonization by bacterial species following FMT can be predicted on the basis of the abundance and phylogeny of the bacteria in the donor and pre-FMT patient samples, with donor strains engrafting in an all-or-nothing manner[60]. Lastly and importantly, the use of FMT instead of ABX for patients with diarrhea should be pursued with caution, given that the present data were obtained from a study of cattle. Its clinical application should be restricted to cattle until more comprehensive data are obtained in other species, including humans.

In conclusion, we have shown that intensive multi-donor FMT is a promising treatment for calf diarrhea and that such gut microbial manipulation could offer another therapeutic paradigm, beyond antibiotic-based therapies. This notion is supported by our findings that there were clear compositional changes in the gut microbiota of diarrheic calves in response to FMT, which were accompanied by alterations in fecal microbial metabolite concentrations. In addition, we have shown that FMT promotes subsequent body mass gain during the fattening stage. This study has also shown that FMT is easier to perform in livestock than human patients, and suggests that it is associated with improvements in both animal welfare and profitability.

## Methods

### Study design and animals

*Study design.* The study was designed to determine the therapeutic efficacy of fecal transplantation from healthy donor calves to recipient's calves at various stages of the progression of diarrhea and to characterize the associated gut microbial dysbiosis or eubiosis. This was assessed by (i) identifying the temporal changes in the calf gut microbiota in response to FMT using metataxonomic analysis, (ii) determining the role of diarrhea-associated dysbiosis in the amelioration of diarrhea in recipient calves transplanted with healthy gut microbiota, (iii) characterizing the gut metabolites that are associated with dysbiosis and the remission of diarrhea using untargeted metabolomics, and (iv) evaluating the ability of FMT to improve growth performance. Korean brown cattle (*Bos taurus coreanae*) were studied. The study protocol was approved by the Institutional Review Boards of the Kyung Hee University [KHUASP(SE)-17-028 and KHUASP(SE)-17-145] and complied with the Animal Protection Act and the Animal Welfare Guidelines of the World Animal Health Organization (WOAH, OIE).

*Animals.* All the calves had free access to food and water, and the mothers nurtured their calves in individual barns. Prior to weaning (60 days after birth), each calf was housed with their mother, and thereafter each mother–calf pair was separated. Although not all the calves were kept in the same space, a physical space was created that could be accessed by the calves, where they could eat their feed and interact with other calves. After weaning (60 days after birth), all the calves were randomly mixed and housed in stalls containing five calves each. All the groups were housed in stalls and fed the same appropriate diet for each stage of growth (Supplementary Table 5) to minimize any stall- or diet-induced inter-individual variation in their intestinal microbiota. All the barns were divided into 3 by 3 m spaces. The floor was kept dry, and the individual buckets and feed bins were cleaned daily throughout the study.

*Preliminary trial.* Seven calves with similar birth dates were studied, irrespective of the presence or severity of diarrhea, to investigate the safety and efficacy of oral FMT (Supplementary Table 1). Out of the seven selected calves, the cleanest calf with the most normal feces, on the basis of the BSS, was selected as the donor. Calves that had been treated with antibiotics or other medications were excluded from the trial. Six recipient calves (three female and three male) were orally administered twice daily with a fecal suspension (40 ml, 0.0005 g/ml feces) containing fecal microbes that had been harvested from the healthy male donor (Supplementary Fig. 1a, b). Then, fecal samples were collected per rectum 0, 1, 2, 4, 8, and 16 days after the treatment. To characterize the calves' environment, we also sampled the feed pellets ($n = 2$) and maternal milk ($n = 6$).

*Validation trial.* To confirm the effects of FMT on calf diarrhea, we used larger numbers of donors and recipients in a second trial (Supplementary Tables 2–4). Calves with similar birth dates were randomly allocated to treatment groups, regardless of sex. Of the 20 healthy calves that were to provide fecal samples, only six (two females and four males) fulfilled the rigorous inclusion and exclusion criteria. Fifty-seven calves with current moderate-to-severe diarrhea (BSS for liquidity 6–7) were studied. The diarrheic calves were orally administered fecal microbiota (FMT, $n = 20$), saline buffer (CON, $n = 14$), or antibiotics (ABX, $n = 23$). The FMT group received 5 g feces as a bolus (50 ml, 0.1 g/ml) on five occasions (0, 12, and 24 h, and 8 and 48 days after the start of the trial). Age-matched diarrheic calves were treated with saline or antibiotics as negative and treatment

controls, respectively. The calves in the CON group were administered saline (50 ml) at the same times as the FMT group were treated, and the calves in the ABX group were orally administered neomycin at the same time points. In the ABX group, if diarrhea recurred, apramycin, sulfadiazine, trimethoprim, or enrofloxacin were injected intramuscularly. Further details of the treatments are given below.

## FMT experiments

*Healthy fecal donors.* Twenty healthy calves aged 21–50 days were selected on the basis of their physical condition (for example, if they had clean noses, mouths, ears, rump, tail, and hair) and nutritional status. Calves that had been administered antimicrobial agents (antibiotics, antifungals, or antivirals) or that had a history of an infectious disease (for example, foot-and-mouth disease, contagious bovine pleuropneumonia, anthrax, or brucellosis) were excluded. Calves with a BSS for feces collected by rectal enema of 3–4 were considered to be healthy and suitable for use as fecal donors. Specific pathogens were tested for in fecal samples using a diagnostic multiplex polymerase chain reaction (PCR) assay, and apparently healthy calves that were positive for causative viral agents (group A rotavirus, group B rotavirus, group C rotavirus, bovine coronavirus, bovine torovirus, bovine norovirus, bovine enteric Nebraska-like calicivirus, bovine nebovirus, or bovine viral diarrhea virus), bacterial agents (*Clostridium perfringens*, *Salmonella enterica*, *S. enterica typhimurium*, enterotoxigenic *E. coli*, shigatoxigenic *E. coli*, or enterohemorrhagic *E. coli*), or protozoal agent (*E. zuernii*) were also excluded (14 calves; Supplementary Table 6). The inclusion and exclusion criteria for healthy fecal donor calves are listed in Table 1.

*Recipient calves with diarrhea.* Thirty-four calves aged 5–50 days that had moderate-to-severe diarrhea (BSS liquidity 6–7) were selected for FMT administration, but 14 calves that had previously been administered antimicrobial agents or had a history of an infectious disease were excluded. The inclusion and exclusion criteria for the recipient's calves are also listed in Table 1.

*FMT procedure.* We collected feces per rectum from healthy suckling calves. A 5 g fecal sample for each calf was homogenized and diluted in 45 ml buffer composed of 4% saline [saline buffer: 4.68% sodium citrate (w/v), 2.29% sodium propionate (w/v), 3.92% sodium acetate (w/v), 5.59% sodium chloride (w/v), 3.56% potassium chloride (w/v), 75.2% glucose (w/v), 0.1% sunset yellow (w/v), 3.0% silicon dioxide (w/v), and 1.62% potassium dihydrogen orthophosphate (w/v)] and 10% glycerol as a cryoprotectant. Each fecal inoculum was prepared to a final concentration of 0.1 g/ml and frozen if it is met healthy fecal donor selection criteria. The control inoculum was isotonic saline. We used a water bath to thaw the frozen fecal inoculums 1 hour before administration and then performed multi-donor investigational infusions by mixing three-to-four fecal donor samples to increase the microbial diversity of the inoculum. We randomly selected the number and identity of the donors for each batch, on the basis of availability.

## Sample collection

*Collection of feces from the rectum.* Fecal samples were collected by rectal enema while wearing clean disposable latex gloves. In the preliminary trial, fecal samples were collected 0, 1, 2, 4, 8, and 16 days after FMT. A total of 37 (donor calf, $n = 1$; recipient calves, $n = 36$) fresh fecal samples were used for the metataxonomic analysis. In the validation trial, fecal samples were collected 0, 2, 4, 8, 16, 32, and 48 days, and 12 months after treatment. A total of 450 (CON, $n = 108$; ABX, $n = 176$; FMT, $n = 166$) fresh fecal samples were used for the metataxonomic analysis. The collected samples were transported to the laboratory on dry ice and stored at −80 °C until use.

*Serum.* Blood samples (5 ml) were collected from the jugular vein by a veterinarian and immediately centrifuged in Microtainer chemistry tubes (Becton Dickinson, Franklin Lakes, NJ, USA) to obtain serum. To reduce the bias associated with the time of sampling, the blood samples for serum metabolite measurements were collected by the same person at a predetermined time (after the cattle had finished their meal). A total of 50 serum samples were transported to the laboratory on dry ice and stored at −80 °C until use.

*Maternal milk.* In the preliminary trial, milk samples ($n = 6$) were collected from the mothers of recipient's calves. To prevent the milk samples from being contaminated with environmental microbes, the cows' udders and teats were wiped with cotton wool soaked in 70% ethanol, and the first few streams of milk were discarded prior to sample collection. The samples were frozen immediately upon collection and then stored at −80 °C until DNA extraction and sequencing.

## Fecal microbial profiling

*DNA extraction and metataxonomic sequencing.* Bacterial genomic DNA was extracted from the calf feces (487 samples from 63 calves), maternal milk (six samples), and feed pellets (two samples) using a Repeated Bead-Beating plus column (QIAamp DNA stool mini kit; Qiagen, Valencia, CA, USA)[61] (Supplementary Tables 1–4). In preparation for Illumina MiSeq sequencing, a fragment of the 16 S rRNA gene spanning the hypervariable V3–V4 region was amplified by PCR using

the forward primer 338F (5′-TCG GCA GCG TCA GAT GTG TAT AAG AGA CAG CCT ACG GGN GGC WGC AG-3′) and the reverse primer 805R (5′-GTC TCG TGG GCT CGG AGA TGT GTA TAA GAG ACA GGA CTA CHV GGG TAT CTA ATC C-3′). PCR was performed in a C1000 Thermal Cycler (Bio-Rad, Hercules, CA, USA). The PCR conditions were as follows: initial denaturation at 95 °C for 3 min; followed by 23 cycles of denaturation at 95 °C for 30 s, annealing at 55 °C for 30 s, and extension at 72 °C for 30 s; and then a final extension step at 72 °C for 5 min. Products of three PCR reactions using the same template were pooled and purified using the QIAquick PCR purification kit (Qiagen). For the Illumina MiSeq sequencing, a 16S V3–V4 PCR product library was prepared using the Nextera XT Index (Illumina). The library was then sequenced on the Illumina MiSeq platform and a paired-end $2 \times 300$ bp reagent kit, according to the manufacturer's instructions.

*Evaluation of DNA contamination.* We assessed all the reagents used for DNA extraction for possible DNA contamination. PCR analysis targeting the hypervariable V3–V4 region of the 16S rRNA gene (30 cycle reaction) revealed no apparent contamination of any of the reagents used. PCR amplicons prepared from DNA extracted from a standard microbial community (ZymoBIOMICS™ Microbial Community Standard; Zymo, Irvine, CA, USA; $n = 2$) and DNA extraction/PCR controls (PCR products of a template acquired from a sham extraction to which no fecal sample had been added; $n = 2$) were included in each MiSeq run for quality control purposes. For taxonomic annotation of the standard community (positive control), a representative sequence for each ASV was aligned with the sequences in the SILVA 123 QIIME-compatible database.

*Sequencing data analysis.* The adapter sequences were trimmed from the raw fastq files, and the trimmed reads were demultiplexed according to the samples using the bcl2fastq2 conversion software version 2.20.0. (Illumina). The sorted reads were imported and processed using QIIME2 version 2020.06[62] for further bioinformatics analyses. The imported paired reads were quality filtered, denoised, and merged using the plugin DADA2 (version 3.11)[63] to generate the ASV feature table. Chimeric sequences and singleton ASVs were excluded from further analyses. Taxonomic classification was performed using the plugin q2-feature-classifier[64], a taxonomic classifier plugin for the QIIME 2 microbiome analysis platform (https://qiime2.org/), which makes similar calculations using a scikit-learn[65] naive Bayes classifier. Finally, taxonomy was assigned to filtered ASVs using a pretrained QIIME2-compatible SILVA version 132 database[66], with 99% identity for the bacteria and representative sequences. To determine the species diversity in each sample, alpha and beta diversity analyses were performed using the plugin q2-diversity in QIIME2 version 2020.06, at a sampling depth of 28,147 for the preliminary trial set, 20,125 for the validation trial set, and 20,293 for the 12-month-old cattle set. Overviews of the Illumina-Miseq-generated datasets are provided in the supplementary tables as follows: for the preliminary study (54 samples), Supplementary Data 1; for the validation study (450 samples), Supplementary Data 2 and 3. All 16 S rRNA gene sequencing data have been deposited in the European Nucleotide Archive and are available under the accession number PRJEB35993. To compare bacterial communities between individuals and between groups, unweighted and weighted UniFrac outputs were assessed and visualized using unsupervised PCoA and distance histograms. Differences between groups were identified using PERMANOVA. Treatment-dependent features were identified using LEfSe 1.0 (Biobakery/Vagrant VM)[67]. A size-effect threshold of 3.5 on the logarithmic LDA score was used to identify discriminating taxa. The Mann–Whitney $U$ test was used to identify taxa that were significantly different in their relative abundance among the groups. Details related to the analysis are described through Code availability.

*Multiple variable correlation analysis.* Correlation analysis was used to identify the strengths of the relationships between the relative abundance of the family *Enterobacteriaceae* or the family *Porphyromonadaceae* and the incidence of diarrhea in intermittently diarrheic calves. Pearson correlation coefficients and statistical significance (two-tailed) for each pair of variables were calculated using data from individual calves or merged data. Spearman's correlation analysis was used to identify associations with the bacterial taxa, with no specified statistical threshold being set.

*SourceTracker analysis.* The tool SourceTracker (version 1.0.1) was used to estimate the proportions of microbes from each source in a target microbial community (sink)[68]. In the preliminary trial analysis, all the recipient samples were entered as individual sinks, while the donor feces, food pellets, and maternal milk were entered as individual sources. In the validation trial analysis, all the recipient samples were entered as individual sinks, while all the donor feces were entered as individual sources. The ASV feature tables and mapping files corresponding to each analysis set were used as input files, and the SourceTracker analysis was performed using default settings. Details related to the analysis are described through Code availability.

*PICRUSt analysis.* PICRUSt 1.1.4 (http://picrust.github.io)[35], for predicting gene families based on the 16 S rRNA gene composition, was used to characterize the functional profiles of the 12-month-old cattle gut microbial community. The

constructed ASV feature table was converted into the PICRUSt format and normalized to 16S rRNA gene copy number to correct for over- and under-estimation of microbial abundance. The normalized dataset was analyzed using the KO (Kyoto Encyclopedia of Genes and Genomes orthology) dataset[69].

**Diagnostic multiplex PCR assay**. To detec the presence of RNA viruses, bacterial genes, and protozoal species, both RNA and DNA were isolated from the calf fecal samples. The isolation methods were as described above. A multiplex PCR assay was performed using previously published primer sequences (Supplementary Table 7). The PCR conditions for the detection of RNA viruses were as follows: initial denaturation at 94 °C for 2 min; followed by 30 cycles of denaturation at 94 °C for 30 s, annealing at 60 °C for 30 s, and extension at 72 °C for 20 s. The PCR conditions for the detection of bacterial genes were as follows: initial denaturation at 94 °C for 2 min; followed by 30 cycles of denaturation at 94 °C for 40 s, annealing at 62 °C for 50 s, and extension at 72 °C for 50 s. A final extension step of 5 min at 72 °C was performed for both reaction types. The PCR conditions for the detection of the internal transcribed spacer 1 (ITS-1) region of the ribosomal RNA gene of bovine *E. zuernii* were as follows: initial denaturation at 94 °C for 30 s; followed by 35 cycles of denaturation at 94 °C for 10 s, annealing at 55 °C for 20 s, and extension at 72 °C for 20 s; and then a final extension at 72 °C for 2 min. All the PCR reactions were performed using a C1000 Thermal Cycler (Bio-Rad, Hercules, CA, USA).

## Metabolomics

*Chemicals*. Methanol and water were purchased from Fisher Scientific (Pittsburgh, PA, USA). Methoxyamine hydrochloride, N-methyl-N-(trimethylsilyl)-trifluoroacetamide, pyridine, and other standard compounds were purchased from Sigma-Aldrich Chemical Co. (St. Louis, MO, USA).

*Sample preparation and metabolite extraction*. Both extracellular and intracellular extracts of fecal samples ($n = 108$) and serum ($n = 50$) were prepared for metabolite profiling, as follows. The rectal luminal contents (250–500 mg) were extracted in 1 ml water using a MM400 mixer mill (Retsch, Haan, Germany) at a frequency of 30 s$^{-1}$ for 5 min at room temperature, followed by sonication for 10 min and incubation at −20 °C for 60 min. After centrifugation (4 °C, 14,764 × g, 10 min), the supernatant was filtered through a polytetrafluoroethylene (0.2 μm) filter and completely dried using a speed-vacuum concentrator (Modulspin 31; Biotron, Korea). For the serum extractions, 1 ml methanol was added to 200 μl serum, and then the mixture was sonicated and shaken for 10 min. After centrifugation (12,580 × g, 4 °C, 10 min), the supernatant was filtered through a 0.2 μm PTFE filter and dried using a speed vacuum concentrator.

*Metabolite profiling and data processing (GC–TOF–MS analysis)*. Dried sample extracts were oximated using methoxyamine hydrochloride (20 mg/ml in pyridine) at 30 °C for 90 min and silylated using N-methyl-N-(trimethylsilyl) trifluoroacetamide at 37 °C for 30 min. The final concentration of the analytes was 10 mg/ml. The analysis was performed using an Agilent 7890 gas chromatography system (Agilent Technologies, Palo Alto, CA, USA), an Agilent 7693 auto-sampler (Agilent Technologies), and a Pegasus HT TOF MS (LECO, St. Joseph, MI, USA) system. One microliter of the sample was injected into the GC–TOF–MS in splitless mode. The analytes were separated on an Rtx-5MS column (30 m I.d. × 0.25 mm length, 0.25 μm particle size; Restek Corp., Bellefonte, PA, USA). Helium gas was used as carrier gas at a constant flow rate of 1.5 mL/min. The temperatures of injector and ion source were set at 250 and 230 °C, respectively. The oven temperature was programmed as follows: 80 °C in the first 2 min, ramped to 300 °C at the rate of 15 °C/min, temperature maintenance for the last 3 min. Analysis of mass range was set 50–1000 *m/z*, with ionization energy of 70 eV.

The GC–TOF–MS data were acquired and pre-processed using LECO Chroma TOF™ software (version 4.44; LECO), and converted to NetCDF format (*.cdf) using LECO Chroma TOF™ software. Then, peak detection, retention time correction, and alignment were performed using MetAlign 3.0 (http://www.metalign.nl), and the data were exported to an Excel file. Multivariate statistical analyses were conducted using the SIMCA-P$^+$ program (version 12.0; Umetrics, Umea, Sweden). The datasets were auto-scaled and mean-centered in a column-wise fashion. The discriminant variables were selected on the basis of the variable importance in projection (VIP) values (VIP > 0.7). Selected metabolites were tentatively identified by comparing mass spectra, retention time, and mass fragment patterns derived from GC–TOF–MS analyses considering standard compounds and in-house library (metabolite identification levels 1 and 2).

To compare metabolite profiles on the basis of differences in concentrations between individuals and groups, Bray–Curtis dissimilarity was assessed and visualized using PCoA and distance histograms. The metabolites that were abundant in the samples were clustered using the UPGMA dendrogram on the basis of the abundance-weighted Jaccard distance (abund_jaccard). The relative concentrations of the metabolites are presented as a heatmap.

## Enzyme assays

*BCAA quantification*. BCAA concentrations were determined in fecal samples (on days 0 and 48 from the three groups of calves ($n = 112$)) and serum samples (6-, 12-, and 24-month-old cattle ($n = 150$)) using a BCAA assay kit (MAK003; Sigma,

St. Louis, MO, USA). All samples and standards were performed in duplicate. Non-drying, moisture-containing fecal samples (10 mg) were homogenized in 4 volumes of cold BCAA Assay buffer. Centrifuge at 13,000 × g for 10 min at 4 °C to remove insoluble material. Serum samples were directly added to the wells. For colorimetric analysis, absorbance was measured at 450 nm (A$_{450}$). A standard curve was constructed for each experiment using a Leucine standard series (MAK003D, Sigma) and used to interpolate the BCAA concentrations.

*Amino acid quantification*. Amino acid concentrations were determined in fecal samples obtained from the calves (on days 0 and 48 from the three groups ($n = 112$)) using an L-Amino Acid quantitation kit (MAK002, Sigma). All samples and standards were assayed in duplicate. Moisture-containing fecal samples (10 mg) were homogenized in four volumes of cold L-Amino Acid Assay buffer, and then the homogenates were centrifuged at 13,000 × g for 10 min at 4 °C to remove insoluble material. The absorbances of the supernatants were measured at 570 nm (A$_{570}$). A standard curve was constructed for each experiment using an L-Amino Acid standard (MAK002E, Sigma), and the unknown values were interpolated.

**Statistics and reproducibility**. All attempts at replication were successful. Also, experiments that were performed in duplicate biologically were described in Methods. We ensured the reproducibility of our findings by performing the same treatment for multiple animals, thus strengthening our results. Statistical analyses were performed using GraphPad Prism version 8.0 for Windows (GraphPad Software, La Jolla, CA, USA). Comparisons between the two samples were made using the Mann–Whitney *U* test (two-tailed). $P < 0.05$ was accepted as representing statistical significance. PERMANOVA was performed to identify the factors determining changes in the calf gut microbiome. Statistical significance for the observed variations was assessed using PERMANOVA, with 999 permutations. The lines, boxes, and whiskers in the box plots represent the median, and 25th, and 75th percentiles, and the min-to-max distribution of replicate values, respectively. The values and scattered dots in the bar graphs represent the means ± SEM and the individual replicates, respectively.

**Reporting summary**. Further information on research design is available in the Nature Research Reporting Summary linked to this article.

## Data availability
The raw sequences of the 16S rRNA genes obtained from the fecal, milk, and feed samples have been deposited in the European Nucleotide Archive, and are available under the accession number PRJEB35993. The mass spectral raw data obtained from the fecal and serum samples have been deposited in the MetaboLights Workbench, and are available under the accession number MTBLS2226. The source data underlying Figs. 1e, 2a–l, 3a–f, 4a–f, 5a–i, and 6a–c, and Supplementary Figs. 1–9 are provided as a Source Data file.

## Code availability
Code used for generating the metataxonomic profiles is publicly available in the Zenodo (https://doi.org/10.5281/zenodo.4176198)[70].

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

## Acknowledgements

We thank the farm owners, veterinarians, and slaughterhouse staff for helping with sample collection. We especially thank Hong Gil Kim (President of the National Hanwoo Association) for his advice regarding the sampling strategy and helpful discussion. This study was supported by the Korea Institute of Planning and Evaluation for Technology in Food, Agriculture, and Forestry (IPET), through the Agricultural Microbiome R&D Program, funded by the Ministry of Agriculture, Food, and Rural Affairs (MAFRA) (grant No. 918011-04-1-SB010); the National Research Foundation of Korea (NRF), funded by the Korean government (MSIT) (grant No. NRF-2018R1A5A1025077); the Mid-Career Researcher Program (grant No. NRF-2020R1A2C3012797); and the Bio & Medical Technology Development Program (grant No. NRF-2017M3A9F3046549).

## Author contributions

J.-W.B., H.S.K., and T.W.W. designed the experiments. H.S.K. and T.W.W. performed the majority of the experiments and analyzed the data. H.S., Y.-S.J., N.-R.S., D.-W.H., P.S.K., and J.-Y.L. helped with sample collection and data presentation. E.S.J. and C.H.L. performed the metabolomics experiments and analyzed the data. H.S.K., T.W.W., and J.-W.B. wrote the paper.

## Competing interests

The authors declare no competing interests.
