## [Peer Review File · Nature Communications]

REVIEWER COMMENTS

Reviewer #1 (Remarks to the Author):

Longitudinal evaluation of fecal microbiota transplantation for ameliorating calf diarrhea and improving growth performance

By Kim et al.

The authors have investigated the effect of providing fecal transplants from healthy donor calves, as a therapy in calves with diarrhea. The authors demonstrate substantial effects on the gut microbiome and metabolome relative to groups receiving either saline or antibiotics. Mortality is markedly reduced and body growth is improved with FMT. To the knowledge of this reviewer, this work is first to show these effects in a cohort of calves with diarrhea. The paper reads really well and is easy to follow. The figures and tables are accurately prepared. The laboratory analytical methods are well described. The authors are congratulated on a job well done. It would be helpful with a better description of their housing conditions and whether all calves were kept in the same facility at the same time. Are there any confounding factors to consider? A better description of diet beyond the milk period would be helpful. When did their rumen function start, and did the authors consider to study the rumen microbiome / metabolome?

minor and major comments:

L58 typo treatment

L163 LEfSe – not clear what this means

L224 relative to what?

L225 only in FMT calves

L233 does the low amino acids concentration in feces from FMT calves indicate improved absorption in the gut?

L254 fig 6 should be fig 5. Have you considered showing these longitudinal data as a graph? and possibly pooling male and female from the notion that there appears to be no sex-specific responses to FMT?

L255 sentence is not meaning full (when full scale of body mass increased...)

L260-263 it would be helpful if you indicate in what way these families discriminate between the three groups

L286 the impact on BCAA at 12 months is impressive. What was the rationale for not measuring this also at 24 months where the body weight differences were more marked ? The serum metabolome represents a snapshot in time and the metabolome might be substantially different, had the sample been taken at another timepoint. Can the authors explain how they tried to standardize the sampling (i.e. time of the day, time after last meal, ect) ?

L305 a reference to this statement is needed

L371-374 it is interesting if the engrafting can really be predicted from the donor or recipient microbiome profiles. This seems key. Can you elaborate to this point ?

L404 and 408 were there seven or six calfs in the safety study ? one of the seven calfs was the donor , - correct ?

L413 please describe the housing conditions. Were these calfs housed and treated in a standardized way ?

L420 what were these criteria ? – ok I see these later in the methods section

L463 typo in this sentence. Also, can you explain why you used this buffer for preservation of the donor material ? why did you pool the donor material from only 3-4 donors ? why not pool all donors ? or alternatively give the un-pooled donor material as this would enable you to study the effects of different donors.

L612 can you elaborate on the amino acids analysis ?

L618 which variables were included in the analysis of variance ? were all calfs housed in the same unit at the same time , - i.e. eliminating block effects ?

Fig 1. The data are impressive. Yet the mortality rates in ABX and CON are surprisingly high, - why is that ? and why did antibiotics not have any effect ? was the diarrhea cases mainly of viral origin?

Fig 4 was this moles of amino acids per mg of dry or liquid feces ?

Reviewer #2 (Remarks to the Author):

Never have I seen pictures of poo tell such a compelling tale. Figure 1 is brilliant and basically tells me all I need to know. The sequencing almost seems like an afterthought (though is of course quite

informative as well, as the other work). There are a few things that could perhaps be improved but I strongly recommend that this smelly piece of work should be accepted (after some discretionary changes).

I mainly have minor comments or (textual) suggestions. I suppose my most important comment is about the interpretation of the fecal concentration of amino acid data (see below).

Abstract:

Here, we found the ability of a fecal
23 microbiota transplantation (FMT), to ameliorate diarrhea and restore gut microbial
24 composition in 57 growing calves.

Replace “found” with “describe”.

We conducted multi-omics analyses of longitudinally
25 collected 450 fecal samples and found that FMT-induced alterations in the gut microbiota (an
26 increase of the family Porphyromonadaceae) and metabolomic profile (a decrease in fecal
27 amino acid concentration) strongly correlated with the remission of diarrhea.

Put “450” in front of “longitudinally”

This first FMT trial for ruminants suggest that the alterations in the
30 gut microbiota may be useful for the treatment of diarrheic calves, and that FMT may have
31 potential role of improvement of growth performance.

Recommend improving English and perhaps making it slightly stronger; the FMT clearly worked.

Recently, however, accumulating evidence has shown that the use of antibiotics
46 in animal husbandry

I would remove the first “,” (Recently however, ...)

58 their use should be reduced in farming and alternatives to antibiotic treatment identified.

Remove the “space” in treatment

172 Facultative anaerobic Gram-negative bacteria (e.g., Proteobacteria) are usually
173 present in small numbers in the gut of healthy adult subjects, where strictly anaerobic bacteria
174 (e.g., Bacteroidetes) tend to predominate²⁸.

I’d say: (e.g., Bacteroidetes and Firmicutes) tend to predominate (See Sup. Fig. 7).

In addition, negative correlations were identified between the relative
198 abundance of the Enterobacteriaceae and that of the Porphyromonadaceae only for the FMT
199 group (CON, Pearson $r = -0.671$, $P = 0.099$; ABX, Pearson $r = -0.215$, $P = 0.643$; FMT,
200 Pearson $r = -0.832$, $P = 0.020$).

“a” negative correlation “was” identified ...

These results indicate the maturation of the gut microbiota
201 following FMT, characterized by an increase in the abundance of the Porphyromonadaceae,
202 which restricts the expansion of the Enterobacteriaceae population and is associated with
203 diarrheic remission.

I think the Porphyromonadaceae are just one of several. Other bacterial families like the Ruminococcaceae, Lachnospiraceae and Bacteroidaceae likely also play a big role. What happens when you combine these 3 abundant families together and then run a comparison?

220 The fecal metabolites in the post-treatment calves were clustered using a Bray–Curtis
221 dissimilarity matrix-based UPGMA dendrogram, and the relative quantities of metabolites
222 were visualized using a heatmap (Fig. 4c). The concentrations of several amino acids
223 (alanine, leucine, valine, isoleucine, glycine, arginine, ornithine, and glutamic acid) were
224 relatively low in the day 48 FMT calves (Supplementary Fig. 5). Indeed, these relative
225 amounts had significantly decreased only FMT calves, from baseline before the 48 day time
226 point (Fig. 4d). By contrast, there were relatively higher concentrations of several amino
227 acids (alanine, valine, isoleucine, and glycine) 48 days after treatment commenced in the
228 CON and ABX groups. Therefore, we quantified fecal amino acid and branched-chain amino
229 acid (BCAA; isoleucine, leucine, and valine) concentrations using an enzyme-based assay,
230 and found that the concentrations of both amino acids (Fig. 4e) and BCAAs (Fig. 4f) were
231 significantly lower in the FMT calves than in the CON and ABX calves at day 48. Taken
232 together, these results suggest that the remission of diarrhea in calves that undergo FMT is
233 accompanied by changes in their metabolomes, and in particular by decreases in amino acid
234 concentrations, which are concomitant with changes in the gut microbiome.

The lower amount of amino acids (branched or not) is the result of achieving a more complete fermentation of the more easily fermentable substrates. Typically, bacteria only really start “eating” the amino acids once there is nothing (or hardly anything else) left (there is more energy to be had by fermenting up sugars or sugar-polymers of various kinds). In a healthy bacterial composition you’ll have a trophic networks of bacteria that is quite efficient in fermenting most things which as a result subsequently leads to that amino acids are also “eaten”. In a dysbiotic gut microbiota composition this trophic network is gone (or disturbed) and as a result fermentation efficiency is down the drain (low). In such a situation there are still enough other sources of energy available (so amino acids remain more untouched). You see exactly the same thing in critically ill infants: more amino acids in the feces of the ill ones, less in the healthy ones; I’d recommend reading and citing “Multi-Compartment Profiling of Bacterial and Host Metabolites Identifies Intestinal Dysbiosis and Its Functional Consequences in the Critically Ill Child”. I’d think this would help for a better interpretation (the data is great and logical). I’d probably recommend rewriting lines 329-356 (quite) a bit with the above in mind.

Here, we have shown that FMT reduces the incidence of diarrhea and changes

306 gut environmental condition from dysbiotic status to eubiotic status in pre-weaning diarrheic
307 calves.

Add "s" to condition(s) and add "a" in front of dysbiotic and eubiotic. (I think...)

These findings

326 collectively suggest that regulating the quantitative changes of the family

327 Porphyromonadaceae in the intestines of young calves may be a cornerstone for resolving of

328 calf diarrhea.

Again, I think that the Porphyromonadaceae are just one of several families that are important (Ruminococcaceae being another obvious candidate). Porphyromonadaceae do not live in a vacuum. They (likely) collaborate with various other species in a trophic network that needs to be built up for optimal health.

357 The present study is the first to use FMT to treat calf diarrhea in place of antibiotics.

358 However, for FMT to be used clinically as a treatment of calf diarrhea, it is necessary to more

359 fully characterize its effects in the gut⁵⁴. First, the underlying microbiological mechanisms

360 whereby members of the family Porphyromonadaceae cause the remission of calf diarrhea

361 must be more fully understood.

If it works it works right? These are not humans we're talking about so you're allowed some risks (in case for some strange unforeseen reason things do not go in the same way as they went in this study). I'm furthermore not even certain that the Porphyromonadaceae are truly the essential link, or more precisely, the only link (though it certainly warrants more investigation).

367 and donors and recipients should be matched on the basis of their microbial profiles, which

would permit customized FMT or defined microbial 368 manipulations.

Again these are not humans and even in humans this kind of “matching” is still just a dream (and possibly even a wrong direction of thinking). If a gut microbiome is severely messed up (diarrhea) you simply need to reconstitute a working fermentative trophic network into the gut (preferably replacing whatever crap was in there previously). In the case of patients where there is just a slight degree of dysbiosis (but not a complete diarrheic mess) I suppose there might be more of a chance of that certain donors will not be successful in getting incorporated into the gut microbiome of the recipient (and that some others will have more success). In any case, when dealing with diarrheic calves, you’re just dealing with a mess (Figure 1 certainly gives that impression) and you simply need healthy calf donors (as you had here).

You might nonetheless in a larger study however certainly want to do more research which kinds of donors have the most success (if any big differences can be found). Perhaps some donors are indeed simply better overall than other donors (like in humans).

This study has also shown that FMT is easier to perform in

382 livestock than human patients, and suggests that it is associated with improvements in both

383 animal welfare and profitability.

Indeed, so I would do away with too much caution. Not only for these reasons, but also for reasons of using less antibiotics in livestock (in regards to antibiotic resistance development; very important in regards to human health) but also for learning FMT lessons that might also be applied in humans.

544 Multiple variable correlation analysis

545 Correlation analysis was used to identify the strength of the relationships between the relative

546 abundance of the family Enterobacteriaceae or the family Porphyromonadaceae and the

547 incidence of diarrhea in intermittently diarrheic calves. Pearson correlation coefficients and

548 statistical significance (two-tailed) for each pair of variables were calculated using data from

549 individual calves or merged data. Spearman’s correlation analysis was used to identify

550 associations with the bacterial taxa, with no specified statistical threshold.

With spearman rho correlation analyses I’m actually typically more interested in the correlation coefficients than in the p-values (as the p-value is dependent on the number of samples while the

correlation coefficient is not so much affected). It's a good thing that in Figure 3 in the mini-heatmaps that the correlation coefficients are indeed given instead of p-values.

In regards to the bioinformatics pipeline, I would recommend perhaps switching to Dada2 or perhaps Mothur-Oligotyping-ARB, looking at ASVs or Oligotypes and then giving representative sequences of these ASVs or Oligotypes (so people can check the species themselves with their own taxonomic identification software if they are interested). This might provide a bit more detail than the current QIIME pipeline can provide. Also, QIIME2 now exists (so at least (eventually) switch to that). However, for the present study the current bioinformatics pipeline in general provided more than enough resolution as the results were so abundantly clear. Figure 5F could however have been more informative with more resolution.

Best regards,

Marcus de Goffau

Responses to Reviewer's comments

(Longitudinal evaluation of fecal microbiota transplantation for ameliorating calf diarrhea
and improving growth performance)

(The authors' replies are written in blue)

Manuscript Number: NCOMMS-20-21979

Comments from reviewer(s):

Referee #1 (Remarks to the Author):

Longitudinal evaluation of fecal microbiota transplantation for ameliorating calf diarrhea and improving growth performance

By Kim et al.

The authors have investigated the effect of providing fecal transplants from healthy donor calves, as a therapy in calves with diarrhea. The authors demonstrate substantial effects on the gut microbiome and metabolome relative to groups receiving either saline or antibiotics. Mortality is markedly reduced and body growth is improved with FMT. To the knowledge of this reviewer, this work is first to show these effects in a cohort of calves with diarrhea. The paper reads really well and is easy to follow. The figures and tables are accurately prepared. The laboratory analytical methods are well described. The authors are congratulated on a job well done.

Response: Thank you for your critical review and valuable comments. As requested, we have performed additional experiments and included the data in the revised manuscript. Our responses are set out below.

1. It would be helpful with a better description of their housing conditions and whether all calves were kept in the same facility at the same time. Are there any confounding factors to consider? A better description of diet beyond the milk period would be helpful.

Response: As per your comment, we have added information regarding the diets to the revised manuscript as Supplementary Table 3.

Supplementary Table 3 The composition of the diets

Dietary component	Ingredients	Proportion (%)			
		Calf (2–6 months)	Juvenile (7–17 months)	Postpubescent (18–24 months)	Adult mother
Solid feed	Corn	-	5	15	4
	Concentrated feed	10	10	10	10
	Wheat	10	10	10	10
	Brewers' grain	10	10	10	10
	Corn Gluten Feed	-	5	6	5
	Wheat gluten	-	5	5	5
	Rice bran	-	4	5	4
	Soybean curd residue	3	5	5	5
	Oil cake	2	4	4	6
	Soybean meal	5	5	2	-
Solid feed total		40	63	72	75
Roughage	Timothy	25	11	-	-
	Alfalfa	30	5	-	-
	Sudan grass	-	10	10	15
	Annual Ryegrass	-	-	7	16
	Roughage total		55	26	17
Others	Water	4.7	10.8	10.9	9.8
	Vitamin (PK-BV3)	0.3	0.2	0.1	0.2
		100	100	100	100

As you have suggested, the details of the animal experiments, including the housing conditions and dietary information, are now described in more detail in the Methods section of the revised manuscript, as follows:

Line 435: “*Animals*

All the calves had free access to food and water, and the mothers nurtured their calves in individual barns. Prior to weaning (60 days after birth), each calf was housed with their mother, and thereafter each mother–calf pair was separated. Although not all the calves were kept in the same space, a physical space was created that could be accessed by the calves, where they could eat their feed and interact with other calves. All the groups were housed in

stalls and fed the same appropriate diet for each stage of growth (Supplementary Table 3) to minimize any stall- or diet-induced inter-individual variation in their intestinal microbiota. All the barns were divided into 3 m by 3 m spaces. The floor was kept dry, and the individual buckets and feed bins were cleaned daily throughout the experiment.”

2. When did their rumen function start, and did the authors consider to study the rumen microbiome / metabolome ?

Response: All the calves used in this study were ≤ 30 days of age. The rumen does not function in young calves because the esophageal groove sends liquid food directly to the abomasum, without going through the rumen. A study by Kaba *et al.* showed that this function of the esophageal groove in calves of 3 weeks of age prevents ruminal development, such that these calves cannot be considered as ruminants at this age. Rumen function begins to develop to some extent 60 days after birth; therefore, the period during which we administered liquids to the calves was prior to full ruminal development. As a consequence, we did not consider analyzing the rumen microbiome or metabolome.

Kaba, T., Abera, B. & Kassa, T. Esophageal groove dysfunction: a cause of ruminal bloat in newborn calves. *BMC Vet Res* **14**, 276 (2018). <https://doi.org/10.1186/s12917-018-1573-2>

Minor and major comments:

L58 typo t treatment

Response: We have modified the sentence to read as follows:

Line 56: “Given that antibiotics also have detrimental impacts on ecology and food safety, their use should be reduced in farming and alternatives to antibiotic treatment identified.”

L163 LEfSe – not clear what this means

Response: We have clarified the meaning, as follows:

Line 172: “Linear discriminant analysis of the day 48 metataxonomic data, coupled with effect size measurements (LEfSe), was used to generate a circular cladogram, which indicates that the phyla Verrucomicrobia, Proteobacteria, and Bacteroidetes were the discriminatory taxa of the CON, ABX, and FMT calves (Fig. 3a).”

L224 relative to what ?

Response: We have modified the sentence to read as follows:

Line 236: “The concentrations of several amino acids (alanine, leucine, valine, isoleucine, glycine, arginine, ornithine, and glutamic acid) were relatively low in the day 48 FMT calves when compared with those in the day 48 CON and ABX calves (Supplementary Fig. 6).”

L225 only in FMT calves

Response: We have modified the sentence to read as follows:

Line 239: “Indeed, these relative amounts had significantly decreased from baseline only in FMT calves before the 48 day time point (Fig. 4d).”

L233 does the low amino acids concentration in feces from FMT calves indicate improved absorption in the gut ?

Response: Our data indicate that the fecal concentrations of amino acids (alanine, leucine, valine, isoleucine, glycine, arginine, ornithine, and glutamic acid) decrease as a result of FMT (Fig. 4). Lower fecal concentrations of amino acids are the result of achieving a more complete fermentation of a more easily fermentable substrate. Similar results were obtained in the FMT calves on day 48, which implies the presence of a relatively healthy gut microbial community, compared with that of CON and ABX calves at the same time point. We have modified the Discussion to clarify this point as follows:

Line 359: “Gut bacteria can use amino acids to synthesize proteins and other metabolites, which play important roles in the nutrition and physiology of the host⁴⁷⁻⁴⁹. Studies conducted in mice have shown that gut bacteria alter the distribution of free amino acids in the gastrointestinal tract and affect the bioavailability of amino acids for the host⁵⁰. When the gut microbiota is healthy (eubiotic status), a nutritional network is maintained that is suitable for fermentation of substances present in the intestine, but when the gut microbial balance is disrupted (dysbiotic status), a nutritional network with low fermentation efficiency is maintained⁵¹.”

47. Cani, P.D. et al. Microbial regulation of organismal energy homeostasis. *Nat Metab* **1**, 34-46 (2019).

48. Dodd, D. et al. A gut bacterial pathway metabolizes aromatic amino acids into nine circulating metabolites. *Nature* **551**, 648-652 (2017).

49. Sonnenburg, J.L. & Backhed, F. Diet-microbiota interactions as moderators of human metabolism. *Nature* **535**, 56-64 (2016).

50. Macfarlane, G.T., Allison, C., Gibson, S.A. & Cummings, J.H. Contribution of the microflora to proteolysis in the human large intestine. *J Appl Bacteriol* **64**, 37-46 (1988).
51. Wijeyesekera, A. et al. Multi-Compartment Profiling of Bacterial and Host Metabolites Identifies Intestinal Dysbiosis and Its Functional Consequences in the Critically Ill Child. *Crit Care Med* **47**, e727-e734 (2019).

L254 fig 6 should be fig 5. Have you considered showing these longitudinal data as a graph ? and possibly pooling male and female from the notion that there appears to be no sex-specific responses to FMT ?

Response: The longitudinal data for the body mass gain were already presented as Supplementary Fig. 6 in the original manuscript (Supplementary Fig. 7 in the revised manuscript). The difference in body mass, according to sex, was significant at 6 months when the body masses of cattle were first measured. Also, because the sex ratios for each group were different, this had the potential to confound the interpretation of the data. Therefore, we presented the female and male body masses separately to minimize the effect of sex and to more clearly show the differences in body masses for each group. However, as shown in the figure below, even when the male and female body masses are analyzed together, the same result is obtained. Therefore, we choose to retain the existing format of the data because this provides more information.

L255 sentence is not meaning full (when full scale of body mass increased...)

Response: We have modified the sentence to read:

Line 269: “We created metataxonomic profiles for the gut microbiota in the 12-month-old cattle ($n = 51$) in which the overall body mass was increased by FMT.”

L260-263 it would be helpful if you indicate in what way these families discriminate between the three groups

Response: To determine the effect of dominant taxa in the gut of CON, ABX, and FMT cattle, we performed PICRUSt (Phylogenetic Investigation of Communities by Reconstruction of Unobserved States) analysis to identify different functional profiles. The gene families predicted using PICRUSt are associated with the enrichment of features required for fast growth (e.g., ribosome, DNA repair and recombination proteins, and DNA

replication proteins) and BCAAs biosynthesis (valine, leucine, and isoleucine biosynthesis) in the FMT cattle (Supplementary Fig. 8b). In summary, the microbial groups associated with FMT may be able to respond more flexibly to dietary intervention at the fattening stage. Information relating to the PICRUSt results is described in the revised manuscript in lines 284–291 and in the Methods section (lines 609–614).

“Supplementary Fig. 8 Relative abundances of the intestinal microbiota in 12-month-old CON, ABX, and FMT cattle

b The predicted functional abundance of the rectal microbiota in 12-month-old CON, ABX, and FMT cattle, as determined using LEfSe and PICRUSt. The abundances are represented by LDA scores (>2.0). CON, control; ABX, antibiotic; FMT, fecal microbiota transplantation. Related to Fig. 5e–i.”

Line 285: “Subsequently, a PICRUSt (Phylogenetic Investigation of Communities by Reconstruction of Unobserved States) pipeline³⁵ was used to determine whether different microbial groups in the three groups might be associated with differences in functional profiles. The gene families predicted using PICRUSt revealed enrichment of features required for fast growth (e.g., ribosome, DNA repair and recombination proteins, and DNA replication proteins) and BCAA biosynthesis (valine, leucine, and isoleucine biosynthesis) in the FMT

cattle (Supplementary Fig. 8b). In summary, various microbial groups formed through FMT may be able to respond more flexibly to dietary intervention during fattening.”

Line 612: “*PICRUSt analysis*

PICRUSt (<http://picrust.github.io>)³⁵, for predicting gene families based on 16S rRNA gene composition, was used to characterize the functional profiles of the 12-month-old cattle gut microbial community. The constructed ASV feature table was converted into the PICRUSt format and normalized to 16S rRNA gene copy number to correct for over- and under-estimation of microbial abundance. The normalized dataset was analyzed using the KO (Kyoto Encyclopedia of Genes and Genomes orthology) dataset⁷¹.”

35. Langille, M.G.I. et al. Predictive functional profiling of microbial communities using 16S rRNA marker gene sequences. *Nat Biotechnol* **31**, 814-821 (2013).

71. Kanehisa, M. & Goto, S. KEGG: Kyoto Encyclopedia of Genes and Genomes. *Nucleic Acids Res* **28**, 27-30 (2000).

L286 the impact on BCAA at 12 months is impressive. What was the rationale for not measuring this also at 24 months where the body weight differences were more marked ? The serum metabolome represents a snapshot in time and the metabolome might be substantially different, had the sample been taken at another timepoint. Can the authors explain how they tried to standardize the sampling (i.e. time of the day, time after last meal, ect) ?

Response: Thank you for your valuable comments. As can be seen in Fig. 5a–d and Supplementary Fig. 7, the differences in body mass between the three groups began to appear at 12 months, so we thought that this represented the time when physiological changes induced by FMT had become apparent. However, we agree with your comments, so we have

now also measured the concentrations of BCAAs in serum samples of 24-month-old-cattle. Consistent with the results obtained in the 12-month-old cattle, the concentrations of BCAAs in the 24-month-old cattle were significantly higher in the sera of FMT cattle than in the other groups. The concentrations of BCAAs in the serum of 24-month-old cattle are now described in the revised manuscript on line 305 and in Fig. 6c, as follows:

Line 306: “To corroborate the metabolomic data, enzyme assays were performed to determine the concentrations of BCAAs in the sera of 6-, 12-, and 24-month-old cattle. Consistent with the results of the metabolomic analysis, the concentration of BCAAs was significantly higher in the serum of FMT cattle than in the other groups, regardless of their age (Fig. 6c).”

“Fig. 6 Serum metabolome profiles of the 12-month-old cattle

c Serum branched-chain amino acid (BCAA) concentrations in the 6- (upper), 12- (middle), and 24- (lower) month-old cattle, quantified enzymatically and displayed as box and dot plots,

as described in the Materials and Methods section. Data are shown as mean \pm SEM and were analyzed using the Mann–Whitney U-test (two-tailed; * $P < 0.05$, ** $P < 0.01$, and *** $P < 0.001$).”

As you have pointed out, the sampling time is very important because serum metabolite concentrations significantly change with time. To reduce the bias associated with the time of sampling, all the blood samples obtained for serum metabolite measurements were collected by the same person at a predetermined time (after the cattle had finished their meal). This sampling time is now described in the revised Methods section, as follows:

Line 522: “*Serum*

Blood samples (5 ml) were collected from the jugular vein by a veterinarian and immediately centrifuged in Microtainer chemistry tubes (Becton Dickinson, Franklin Lakes, NJ, USA) to obtain serum. To reduce the bias associated with the time of sampling, the blood samples for serum metabolite measurements were collected by the same person at a predetermined time (after the cattle had finished their meal). A total of 50 serum samples were transported to the laboratory on dry ice and stored at -80°C until use.”

L305 a reference to this statement is needed

Response: We now cite references to support this statement, as follows:

Line 327: “Unfortunately, although attempts have been made to treat gastrointestinal diseases by administering livestock a fecal inoculum or specific bacterial species, no previous studies have shown meaningful changes in the gut microbial community^{37,38}.”

37. Hu, J. et al. A Microbiota-Derived Bacteriocin Targets the Host to Confer Diarrhea Resistance in Early-Weaned Piglets. *Cell Host Microbe* **24**, 817-832 (2018).

38. Brunse, A. et al. Effect of fecal microbiota transplantation route of administration on gut colonization and host response in preterm pigs. *Isme J* **13**, 720-733 (2019).

L371-374 it is interesting if the engrafting can really be predicted from the donor or recipient microbiome profiles. This seems key. Can you elaborate to this point ?

Response: To provide an accurate basis for the answer to the above question, we performed SourceTracker analysis. In the preliminary trial, it was confirmed that donor calf feces contributed significantly to the formation of intestinal microbial communities in the recipient calves, unlike ingested materials (food pellets and maternal milk) (Supplementary Fig. 2).

“Supplementary Fig. 2 Estimation of the contribution of each source of bacteria to the microbial communities of the recipient calves in the preliminary trial

a–b Results of SourceTracker analysis, showing the mean contributions of each source of bacteria to the bacterial communities of the recipient calves **(a)** on day 0 and **(b)** day 16 after the start of treatment. The sources of the bacteria were the gut microbiota of donor feces, food pellets, and maternal milk. Related to Supplementary Fig. 1e.”

In addition, in the validation trial, and unlike in CON or ABX calves, it was confirmed that the feces of the donor calves contributed more significantly to the formation of the intestinal microbial communities of FMT calves (Fig. 2l and Supplementary Fig. 3c, d).

“Supplementary Fig. 3 Contributions of donor feces to the bacterial communities in the CON, ABX, and FMT calves

c–d Results of SourceTracker analysis showing the mean contributions of donor feces to the bacterial communities in the CON, ABX, and FMT calves on **(c)** day 0 and **(d)** day 48. Source: gut microbiome of donor calves; sinks: CON, ABX, and FMT calves. CON, control; ABX, antibiotic; FMT, fecal microbiota transplantation. Related to Fig. 2l.”

Taken together, the results [the dissimilarity values (between donors and the CON or ABX or FMT calves), the number of shared ASVs (between the feces of healthy donor calves and the CON, ABX, and FMT calves), and the results of Source Tracker analysis] imply that the gut microbiota of the donor calves was successfully delivered to the recipient calves. Information relating to the SourceTracker analysis is described in the revised manuscript as follows:

Line 101: “To demonstrate this capability, we used SourceTracker analysis to quantify the proportions of the different microbial samples (sources) in a target microbial community (sink). SourceTracker analysis showed that the mean contributions of the donor feces to the gut bacterial community of the recipient calves increased during the 16 days after the initiation of treatment in all the calves, and that there was no contribution from external sources other than the donor feces (Supplementary Fig. 2a, b).”

Line 160: “SourceTracker analysis was used to determine whether the gut microbiota of CON, ABX, and FMT calves originated from the gut of donor calves during the trial period. The results show that the mean contributions of the donor feces to the gut bacterial community of the CON, ABX, and FMT calves was highest 48 days after the initiation of treatment in FMT calves (Fig. 2l and Supplementary Fig. 3c, d). Thus, FMT reduced inter-individual variation in the microbial community and induced a change in the microbial structure of diarrheic calves toward that of the healthy donor calves.”

Line 602: “*SourceTracker analysis*

The tool SourceTracker was used to estimate the proportions of microbes from each source in a target microbial community (sink)⁷⁰. In the preliminary trial analysis, all the recipient samples were entered as individual sinks, while the donor feces, food pellets, and maternal milk were entered as individual sources. In the validation trial analysis, all the recipient samples were entered as individual sinks, while all the donor feces were entered as individual sources. The ASV feature tables and mapping files corresponding to each analysis set were used as input files, and the SourceTracker analysis was performed using default settings. Details related to the analysis are described through Code availability.”

70. Knights, D. et al. Bayesian community-wide culture-independent microbial source tracking. *Nat Methods* **8**, 761-U107 (2011).

L404 and 408 were there seven or six calfs in the safety study ? one of the seven calfs was the donor , - correct ?

Response: Of the seven selected calves, the cleanest calf with the most normal feces on the basis of the Bristol stool scale (BSS) was selected as the donor. We have added the following text to clarify this:

Line 446: “*Preliminary trial*

Seven calves with similar birth dates were studied, irrespective of the presence or severity of diarrhea, to investigate the safety and efficacy of oral FMT (Supplementary Table 1). Out of the seven selected calves, the cleanest calf with the most normal feces, on the basis of the BSS, was selected as the donor. Calves that had been treated with antibiotics or other medications were excluded from the trial. Six recipient calves (three female and three male) were orally administered twice daily with a fecal suspension (40 ml, 0.0005 g/ml feces) containing fecal microbes that had been harvested from the healthy male donor (Supplementary Fig. 1a,b). Then, fecal samples were collected per rectum 0, 1, 2, 4, 8, and 16 days after the treatment. To characterize the calves’ environment, we also sampled the feed pellets ($n = 2$) and maternal milk ($n = 6$).”

L413 please describe the housing conditions. Were these calfs housed and treated in a standardized way ?

Response: The details of the animal experiments, including the housing conditions and dietary information, are described in more detail in the revised Methods section. We have added information regarding the diets to the revised manuscript as Supplementary Table 3 and to the text as follows:

Line 435: “*Animals*

All the calves had free access to food and water, and the mothers nurtured their calves in individual barns. Prior to weaning (60 days after birth), each calf was housed with their mother, and thereafter each mother–calf pair was separated. Although not all the calves were kept in the same space, a physical space was created that could be accessed by the calves, where they could eat their feed and interact with other calves. All the groups were housed in stalls and fed the same appropriate diet for each stage of growth (Supplementary Table 3) to minimize any stall- or diet-induced inter-individual variation in their intestinal microbiota. All the barns were divided into 3 m by 3 m spaces. The floor was kept dry, and the individual buckets and feed bins were cleaned daily throughout the experiment.”

L420 what were these criteria ? – ok I see these later in the methods section

Response: Thank you for your critical reading. We provide the inclusion and exclusion criteria for the recipient calves in Table 1.

L463 typo in this sentence. Also, can you explain why you used this buffer for preservation of the donor material ? why did you pool the donor material from only 3-4 donors ? why not pool all donors ? or alternatively give the un-pooled donor material as this would enable you to study the effects of different donors.

Response: We could not find a typo error in the sentence (Line 505).

The buffer we used is an electrolyte mixture that prevents dehydration in calves and was administered to the diarrheic calves of the CON group. To protect the donor material, 10% glycerol was added.

The rationale for performing multi-donor fecal microbiota transplantation was that it is an effective method of increasing microbial diversity, as shown in a previous publication, albeit

that the subjects of the previous experiment differed (Multidonor intensive faecal microbiota transplantation for active ulcerative colitis: a randomised placebo-controlled trial. *Lancet* **389**, 1218-1228 (2017)). The authors blended the feces of between three and seven donors to increase microbial diversity. However, when performing multi-donor FMT, the use of more than three donors did not significantly affect the success of FMT. Therefore, in the present study, we believe that pooling material from only three or four donor calves should have been effective.

L612 can you elaborate on the amino acids analysis ?

Response: We have added more details of the method used for the measurement of amino acid concentrations to the Methods section, as follows:

Line 684: “*Amino acid quantification*

Amino acid concentrations were determined in fecal samples obtained from the calves (on days 0 and 48 from the three groups ($n = 112$)) using an L-Amino Acid quantitation kit (MAK002, Sigma). All samples and standards were assayed in duplicate. Moisture-containing fecal samples (10 mg) were homogenized in four volumes of cold L-Amino Acid Assay buffer, and then the homogenates were centrifuged at 12,000 r.p.m. for 10 min at 4°C to remove insoluble material. The absorbances of the supernatants were measured at 570 nm (A_{570}). A standard curve was constructed for each experiment using an L-Amino Acid standard (MAK002E, Sigma), and the unknown values were interpolated.”

L618 which variables were included in the analysis of variance ? were all calfs housed in the same unit at the same time , - i.e. eliminating block effects ?

Response: Cows are raised in an environment that cannot be precisely controlled, unlike in the cases of mice or other experimental animals, and because of these environmental characteristics, FMT trials are conducted outdoors. Therefore, we acknowledge your concerns and comments. Because the calves were fed by their mothers before weaning, they could not all be raised in the same space. The housing incorporated spaces that the calves could access, wherein they could consume feed and interact with other calves. All the groups were held in stalls and fed identical diets for each stage of their growth (Supplementary Table 3) to minimize any cage- and diet-induced inter-individual variations in their intestinal microbiota. All the barns comprised 3 m by 3 m spaces. The floor was kept dry, and the individual buckets and feed bins were cleaned daily throughout the experiment. At the beginning of the experiment, more than 300 fertile cows were present, and the calves were born between April and June of the following year, following artificial insemination. Only calves born during this period were used in the experiment, so that there would be no or little effect of season on the data.

Fig 1. The data are impressive. Yet the mortality rates in ABX and CON are surprisingly high, - why is that ? and why did antibiotics not have any effect ? was the diarrhea cases mainly of viral origin?

Response: Although a limitation of the experimental design was that we could not provide daily BSS data, the ABX group showed a very high incidence of recurrence of diarrhea on days 4–8 after the administration of antibiotic (neomycin). When the diarrhea recurred, apramycin, sulfadiazine, trimethoprim, or enrofloxacin was injected subcutaneously, as described in the Methods section. However, repeated recurrence may not be effectively treated using antibiotics. This is thought to be the result of the problem of the use of

antibiotics that humans continue to worry about. Because they can kill both pathogenic and beneficial microbes, the use of wide-spectrum antibiotics can promote the colonization of the gut by pathogenic microbes, which can cause disease¹⁰⁻¹². Thus, antibiotic therapy of diarrheic calves often leads to the recurrence of serious diarrhea within days of starting it. The presence of an imbalance in the intestinal microbial community, followed by the administration of antibiotics, is capable of activating immune responses, inflammation, and peristalsis in the host gut, which causes diarrhea^{13,14}. Importantly, a combination of recurrent diarrhea and antibiotic abuse in pre-weaning calves may result in immaturity of the ruminal and intestinal microbiota, which has permanent negative effects on the digestion and absorption of dietary components during the fattening period¹⁵⁻¹⁷.

The CON group was administered the electrolyte mixture as a control for the FMT, and this does not represent a definitive treatment for diarrhea, but rather prevents dehydration due to diarrhea. These factors are considered to be the explanation for the high mortality rate in the CON and ABX groups.

10. Sullivan, A., Edlund, C. & Nord, C.E. Effect of antimicrobial agents on the ecological balance of human microflora. *Lancet Infect Dis* **1**, 101-114 (2001).
11. Buffie, C.G. & Pamer, E.G. Microbiota-mediated colonization resistance against intestinal pathogens. *Nat Rev Immunol* **13**, 790-801 (2013).
12. Ubeda, C. & Pamer, E.G. Antibiotics, microbiota, and immune defense. *Trends Immunol* **33**, 459-466 (2012).
13. Spees, A.M., Lopez, C.A., Kingsbury, D.D., Winter, S.E. & Baumler, A.J. Colonization Resistance: Battle of the Bugs or Menage a Trois with the Host? *Plos Pathog* **9** (2013).

14. Rivera-Chavez, F. et al. Depletion of Butyrate-Producing Clostridia from the Gut Microbiota Drives an Aerobic Luminal Expansion of Salmonella. *Cell Host Microbe* **19**, 443-454 (2016).
15. Ji, S. et al. Ecological Restoration of Antibiotic-Disturbed Gastrointestinal Microbiota in Foregut and Hindgut of Cows. *Front Cell Infect Microbiol* **8**, 79 (2018).
16. Wolin, M.J. Fermentation in the rumen and human large intestine. *Science* **213**, 1463-1468 (1981).
17. Russell, J.B. & Rychlik, J.L. Factors that alter rumen microbial ecology. *Science* **292**, 1119-1122 (2001).

Fig 4 was this moles of amino acids per mg of dry or liquid feces ?

Response: Non-drying, moisture-containing feces were homogenized in cold L-Amino Acid Assay buffer. The concentration of amino acids was measured in the supernatant obtained by centrifugation to remove insoluble material. The units are shown in Fig. 4e and below (left y-axis).

“Fig. 4 Changes in the fecal metabolome profile of the diarrheic calves following FMT

e Amino acid concentrations in the fecal samples obtained from calves in each group on days 0 and 48. The amino acid concentrations were measured by an enzymatic method using an L-Amino Acid quantitation kit, and are displayed as box and dot plots.”

Referee #2 (Remarks to the Author):

Never have I seen pictures of poo tell such a compelling tale. Figure 1 is brilliant and basically tells me all I need to know. The sequencing almost seems like an afterthought (though is of course quite informative as well, as the other work). There are a few things that could perhaps be improved but I strongly recommend that this smelly piece of work should be accepted (after some discretionary changes). I mainly have minor comments or (textual) suggestions. I suppose my most important comment is about the interpretation of the fecal concentration of amino acid data (see below).

Response: Thank you for looking carefully at the results of our efforts. As requested, we have performed additional analyses and have included the data in the revised version of the manuscript. Our responses are set out below.

Abstract:

Here, we found the ability of a fecal microbiota transplantation (FMT), to ameliorate diarrhea and restore gut microbial composition in 57 growing calves.

Replace “found” with “describe”.

Response: We have modified the sentence to read:

Line 22: “Here, we describe the ability of fecal microbiota transplantation (FMT) to ameliorate diarrhea and restore gut microbial composition in 57 growing calves.”

We conducted multi-omics analyses of longitudinally collected 450 fecal samples and found that FMT-induced alterations in the gut microbiota (an increase of the family

Porphyromonadaceae) and metabolomic profile (a decrease in fecal amino acid concentration) strongly correlated with the remission of diarrhea.

Put “450” in front of “longitudinally”

Response: We have modified the sentence to read:

Line 24: “We conducted multi-omic analysis of 450 longitudinally collected fecal samples and found that FMT-induced alterations in the gut microbiota (an increase in the family *Porphyromonadaceae*) and metabolomic profile (a reduction in fecal amino acid concentration) strongly correlated with the remission of diarrhea.”

This first FMT trial for ruminants suggest that the alterations in the gut microbiota may be useful for the treatment of diarrheic calves, and that FMT may have potential role of improvement of growth performance.

Recommend improving English and perhaps making it slightly stronger; the FMT clearly worked.

Response: We have modified the sentence to read:

Line 29: “This first FMT trial in ruminants suggest that FMT is capable of ameliorating diarrhea in pre-weaning calves with alterations in their gut microbiota, and that FMT may have potential role of improvement of growth performance.”

Recently, however, accumulating evidence has shown that the use of antibiotics in animal husbandry

I would remove the first “,” (Recently however, ...)

Response: We have removed the first “,” as follows:

Line 45: “Recently however, accumulating evidence has shown that the use of antibiotics in animal husbandry is associated with many adverse effects.”

58 their use should be reduced in farming and alternatives to antibiotic treatment identified.

Remove the “space” in treatment

Response: We have made the change, as follows:

Line 56: “Given that antibiotics also have detrimental impacts on ecology and food safety, their use should be reduced in farming and alternatives to antibiotic treatment identified.”

172 Facultative anaerobic Gram-negative bacteria (e.g., Proteobacteria) are usually present in small numbers in the gut of healthy adult subjects, where strictly anaerobic bacteria (e.g., Bacteroidetes) tend to predominate.

I’d say: (e.g., Bacteroidetes and Firmicutes) tend to predominate (See Sup. Fig. 7).

Response: We have modified the sentence to read:

Line 184: “Facultative anaerobic Gram-negative bacteria (e.g., Proteobacteria) are usually present in small numbers in the gut of healthy adult subjects, where strictly anaerobic bacteria (e.g., Bacteroidetes and Firmicutes) tend to predominate.”

In addition, negative correlation were identified between the relative abundance of the Enterobacteriaceae and that of the Porphyromonadaceae only for the FMT group (CON,

Pearson $r = -0.671$, $P = 0.099$; ABX, Pearson $r = -0.215$, $P = 0.643$; FMT, Pearson $r = -0.832$, $P = 0.020$).

“a” negative correlation “was” identified ...

Response: We have modified the sentence to read:

Line 212: “In addition, a negative correlation was identified between the relative abundance of the *Enterobacteriaceae* and that of the *Porphyromonadaceae* in the FMT group alone (CON, Pearson $r = -0.752$, $P = 0.025$; ABX, Pearson $r = -0.491$, $P = 0.130$; FMT, Pearson $r = -0.824$, $P = 0.024$).”

These results indicate the maturation of the gut microbiota following FMT, characterized by an increase in the abundance of the Porphyromonadaceae, which restricts the expansion of the Enterobacteriaceae population and is associated with diarrheic remission.

I think the Porphyromonadaceae are just one of several. Other bacterial families like the Ruminococcaceae, Lachnospiraceae and Bacteroidaceae likely also play a big role. What happens when you combine these 3 abundant families together and then run a comparison?

Response: We fully agree with your concern. Therefore, we have further analyzed the differences in the abundances of dominant the taxa at the family level (the *Ruminococcaceae*, *Lachnospiraceae*, and *Bacteroidaceae*). As can be seen from the results (Fig. 3e), only the *Enterobacteriaceae* and *Porphyromonadaceae* showed significant differences among the CON, ABX, and FMT groups, and we confirmed a strong correlation with the remission of diarrhea.

“Fig. 3 Discriminating microbial taxa associated with the incidence of diarrhea, and correlation analysis of gut *Enterobacteriaceae* or *Porphyromonadaceae* with the incidence of diarrhea.

e Changes in the relative abundance of the family *Enterobacteriaceae* (left) and *Porphyromonadaceae* (right) with time. Data are shown as mean \pm SEM. *Comparison of the CON and FMT groups; #comparison of the ABX and FMT groups; and †comparison of the CON and ABX groups. * $P < 0.05$, ** $P < 0.01$, and *** $P < 0.001$.”

As a result of simultaneously determining the changes in abundance of the *Ruminococcaceae*, *Lachnospiraceae*, and *Bacteroidaceae* all at once (figure shown below), significantly higher abundance was identified in FMT calves than in CON calves on day 48. This was shown to be due to changes in the abundance of the *Bacteroidaceae*, but there was no correlation with the remission of diarrhea (Bristol stool scale). However, the CON, ABX, and FMT groups did not differ in their populations of *Enterobacteriaceae* or *Porphyromonadaceae*.

The abundances of the families *Ruminococcaceae*, *Bacteroidaceae*, and *Paraprevotellaceae* appear to increase with aging, whereas those of the *Lachnospiraceae* and *Lactobacillaceae* appear to decrease (Supplementary Fig. 5 f–j). The CON, ABX, and FMT groups showed similar populations (*Ruminococcaceae*, *Bacteroidaceae*, *Paraprevotellaceae*, *Lachnospiraceae*, and *Lactobacillaceae*), and there was no strong correlation with the remission of diarrhea; therefore, these taxa are not thought to affect the incidence of diarrhea.

“Supplementary Fig. 5 Changes in the relative abundances of the predominant bacterial phyla and families during the trial

f–j The abundances of the families (f) *Ruminococcaceae*, (g) *Bacteroidaceae*, (h)

Paraprevotellaceae, (i) *Lachnospiraceae*, and (j) *Lactobacillaceae* are shown, according to treatment group. Data are shown as mean \pm SEM. *Comparison of the CON and FMT groups; #comparison of the ABX and FMT groups; and †comparison of the CON and ABX groups. * $P < 0.05$, ** $P < 0.01$, and *** $P < 0.001$. CON, control; ABX, antibiotic; FMT, fecal microbiota transplantation. Related to Fig. 3e.”

220 The fecal metabolites in the post-treatment calves were clustered using a Bray–Curtis dissimilarity matrix-based UPGMA dendrogram, and the relative quantities of metabolites were visualized using a heatmap (Fig. 4c). The concentrations of several amino acids (alanine, leucine, valine, isoleucine, glycine, arginine, ornithine, and glutamic acid) were relatively low in the day 48 FMT calves (Supplementary Fig. 5). Indeed, these relative amounts had significantly decreased only FMT calves, from baseline before the 48 day time point (Fig. 4d). By contrast, there were relatively higher concentrations of several amino acids (alanine, valine, isoleucine, and glycine) 48 days after treatment commenced in the CON and ABX groups. Therefore, we quantified fecal amino acid and branched-chain amino acid (BCAA; isoleucine, leucine, and valine) concentrations using an enzyme-based assay, and found that the concentrations of both amino acids (Fig. 4e) and BCAAs (Fig. 4f) were significantly lower in the FMT calves than in the CON and ABX calves at day 48. Taken together, these results suggest that the remission of diarrhea in calves that undergo FMT is accompanied by changes in their metabolomes, and in particular by decreases in amino acid concentrations, which are concomitant with changes in the gut microbiome.

The lower amount of amino acids (branched or not) is the result of achieving a more complete fermentation of the more easily fermentable substrates. Typically, bacteria only really start “eating” the amino acids once there is nothing (or hardly anything else) left (there

is more energy to be had by fermenting up sugars or sugar-polymers of various kinds). In a healthy bacterial composition you'll have a trophic networks of bacteria that is quite efficient in fermenting most things which as a result subsequently leads to that amino acids are also "eaten". In a dysbiotic gut microbiota composition this trophic network is gone (or disturbed) and as a result fermentation efficiency is down the drain (low). In such a situation there are still enough other sources of energy available (so amino acids remain more untouched). You see exactly the same thing in critically ill infants: more amino acids in the feces of the ill ones, less in the healthy ones; I'd recommend reading and citing "Multi-Compartment Profiling of Bacterial and Host Metabolites Identifies Intestinal Dysbiosis and Its Functional Consequences in the Critically Ill Child". I'd think this would help for a better interpretation (the data is great and logical). I'd probably recommend rewriting lines 329-356 (quite) a bit with the above in mind.

Response: Thank you for your valuable comment. After reading the recommended paper, we have acquired a new perspective regarding the situation where the intestinal amino acid content is low (Multi-Compartment Profiling of Bacterial and Host Metabolites Identifies Intestinal Dysbiosis and Its Functional Consequences in the Critically Ill Child). This is thanks to the clear presentation of the data in this paper and the clear writing style. From this, we understand that a lower concentration of amino acids is the result of more complete fermentation of easily fermentable substrates. We obtained similar results in the FMT calves on day 48, which indicates that they have a healthier gut microbial community than the CON and ABX calves at the same time point. We have amended the Discussion to reflect this. Please refer to the bold text below:

Line 353: "We found that the fecal concentrations of amino acids (alanine, leucine, valine, isoleucine, glycine, arginine, ornithine, and glutamic acid) decrease as a result of FMT (Fig.

4), which suggests that the metabolism of microbes associated with the remission of diarrhea leads to low fecal concentrations of amino acids. Amino acids are necessary for intestinal growth, the maintenance of mucosal integrity, and the barrier function. They are essential precursors for glutathione, polyamines, nitric oxide, and other molecules in intestinal epithelial cells, and are building blocks for the macromolecular synthesis that is necessary for mucosal wound healing and energy production in intestinal cells⁴⁶. **Gut bacteria can use amino acids to synthesize proteins and other metabolites, which play important roles in the nutrition and physiology of the host⁴⁷⁻⁴⁹. Studies conducted in mice have shown that gut bacteria alter the distribution of free amino acids in the gastrointestinal tract and affect the bioavailability of amino acids for the host⁵⁰. When the gut microbiota is healthy (eubiotic status), a nutritional network is maintained that is suitable for fermentation of substances present in the intestine, but when the gut microbial balance is disrupted (dysbiotic status), a nutritional network with low fermentation efficiency is maintained⁵¹. Thus, higher concentrations of amino acids in the intestine are thought to be the result of incomplete fermentation. Similarly, the results of pediatric studies have suggested that a significant fecal loss of amino acids occurs in infant diarrhea: fecal free amino acid concentrations have been shown to be 10-fold higher during diarrhea than during periods of remission or in normal individuals^{52,53}. Furthermore, in a recent study, the dysbiotic gut microbiota of patients with diarrhea was found to be characterized by higher concentrations of free amino acids, especially of proline, and was associated with higher susceptibility to CDI⁵⁴.**

In addition, previous studies conducted in rodents and pigs have provided evidence that certain amino acids, particularly glutamine and arginine, may influence the progression of IBD⁵⁵ and reduce inflammation and oxidative stress. Amino acid supplementation may be beneficial for patients with IBD or cancer, but it may also have adverse effects in the human

bowel. Administration of amino acids such as arginine, cysteine, ornithine, and citrulline causes a variety of gastrointestinal side effects, including nausea, diarrhea, abdominal cramping, and bloating⁵⁶. Furthermore, an amino acid imbalance or an increase in protein catabolism can provoke metabolic acidosis, which is also termed hyperchloremic acidosis, and the most common underlying mechanism involves the loss of large amounts of base because of diarrhea^{57,58}. In light of these previous findings, it is likely that an imbalance in the intestinal amino acids, which is followed by microbial dysbiosis, is associated with a higher risk of diarrhea in calves. Further studies should aim to determine in more detail the effects of amino acid metabolism on gastrointestinal disorders in livestock.”

46. Wu, G. Intestinal mucosal amino acid catabolism. *J Nutr* **128**, 1249-1252 (1998).
47. Cani, P.D. et al. Microbial regulation of organismal energy homeostasis. *Nat Metab* **1**, 34-46 (2019).
48. Dodd, D. et al. A gut bacterial pathway metabolizes aromatic amino acids into nine circulating metabolites. *Nature* **551**, 648-652 (2017).
49. Sonnenburg, J.L. & Backhed, F. Diet-microbiota interactions as moderators of human metabolism. *Nature* **535**, 56-64 (2016).
50. Macfarlane, G.T., Allison, C., Gibson, S.A. & Cummings, J.H. Contribution of the microflora to proteolysis in the human large intestine. *J Appl Bacteriol* **64**, 37-46 (1988).
51. Wijeyesekera, A. et al. Multi-Compartment Profiling of Bacterial and Host Metabolites Identifies Intestinal Dysbiosis and Its Functional Consequences in the Critically Ill Child. *Crit Care Med* **47**, e727-e734 (2019).

52. Ghadimi, H., Kumar, S. & Abaci, F. Endogenous amino acid loss and its significance in infantile diarrhea. *Pediatr Res* **7**, 161-168 (1973).
53. Kolho, K.L., Pessia, A., Jaakkola, T., de Vos, W.M. & Velagapudi, V. Faecal and Serum Metabolomics in Paediatric Inflammatory Bowel Disease. *J Crohns Colitis* **11**, 321-334 (2017).
54. Battaglioli, E.J. et al. Clostridioides difficile uses amino acids associated with gut microbial dysbiosis in a subset of patients with diarrhea. *Sci Transl Med* **10** (2018).
55. Vidal-Lletjos, S. et al. Dietary Protein and Amino Acid Supplementation in Inflammatory Bowel Disease Course: What Impact on the Colonic Mucosa? *Nutrients* **9** (2017).
56. Grimble, G.K. Adverse gastrointestinal effects of arginine and related amino acids. *J Nutr* **137**, 1693S-1701S (2007).
57. Holecek, M., Safranek, R., Rysava, R., Kadlcikova, J. & Sprongl, L. Acute effects of acidosis on protein and amino acid metabolism in perfused rat liver. *Int J Exp Pathol* **84**, 185-190 (2003).
58. Kraut, J.A. & Kurtz, I. Treatment of acute non-anion gap metabolic acidosis. *Clin Kidney J* **8**, 93-99 (2015).

Here, we have shown that FMT reduces the incidence of diarrhea and changes gut environmental condition from dysbiotic status to eubiotic status in pre-weaning diarrheic calves.

Add “s” to condition(s) and add “a” in front of dysbiotic and eubiotic. (I think...)

Response: We have modified the sentence to read:

Line 329: “Here, we have shown that FMT reduces the incidence of diarrhea and changes gut environmental conditions from a dysbiotic status to a eubiotic status in pre-weaning diarrheic calves.”

These findings collectively suggest that regulating the quantitative changes of the family Porphyromonadaceae in the intestines of young calves may be a cornerstone for resolving of calf diarrhea.

Again, I think that the Porphyromonadaceae are just one of several families that are important (Ruminococcaceae being another obvious candidate). Porphyromonadaceae do not live in a vacuum. They (likely) collaborate with various other species in a trophic network that needs to be built up for optimal health.

Response: After treatment, the only positive correlation identified with the remission of diarrhea in FMT calves for the seven dominant taxa was that of the abundance of the family *Porphyromonadaceae* (Fig. 3e and Supplementary Fig. 5). This suggests that the family *Porphyromonadaceae* is associated with the mechanism of diarrheic remission induced by FMT. There are eight genera taxa with a validly published name in the family *Porphyromonadaceae* (<https://lpsn.dsmz.de/family/porphyromonadaceae>), but additional studies are clearly needed because limitations in sequencing resolution may not have facilitated accurate identification at the genus or species level. If bacteria are identified through pure isolation culture in the future, studies of trophic networks must also then be performed. Please see our response to the comment by Reviewer #2 regarding the three most abundant families (*Ruminococcaceae*, *Lachnospiraceae* and *Bacteroidaceae*).

“Fig. 3 Discriminating microbial taxa associated with the incidence of diarrhea, and correlation analysis of gut *Enterobacteriaceae* or *Porphyromonadaceae* with the incidence of diarrhea

e Changes in the relative abundance of the family *Enterobacteriaceae* (left) and *Porphyromonadaceae* (right) with time. Data are shown as mean \pm SEM. *Comparison of the CON and FMT groups; #comparison of the ABX and FMT groups; and †comparison of the CON and ABX groups. * $P < 0.05$, ** $P < 0.01$, and *** $P < 0.001$.”

“Supplementary Fig. 5 Changes in the relative abundances of the predominant bacterial phyla and families during the trial

f–j The abundances of the families (f) *Ruminococcaceae*, (g) *Bacteroidaceae*, (h) *Paraprevotellaceae*, (i) *Lachnospiraceae*, and (j) *Lactobacillaceae* are shown, according to

treatment group. Data are shown as mean \pm SEM. *Comparison of the CON and FMT groups; #comparison of the ABX and FMT groups; and †comparison of the CON and ABX groups. * $P < 0.05$, ** $P < 0.01$, and *** $P < 0.001$. CON, control; ABX, antibiotic; FMT, fecal microbiota transplantation. Related to Fig. 3e.”

357 The present study is the first to use FMT to treat calf diarrhea in place of antibiotics. However, for FMT to be used clinically as a treatment of calf diarrhea, it is necessary to more fully characterize its effects in the gut. First, the underlying microbiological mechanisms whereby members of the family Porphyromonadaceae cause the remission of calf diarrhea must be more fully understood.

If it works it works right? These are not humans we’re talking about so you’re allowed some risks (in case for some strange unforeseen reason things do not go in the same way as they went in this study). I’m furthermore not even certain that the Porphyromonadaceae are truly the essential link, or more precisely, the only link (though it certainly warrants more investigation).

Response: In response to your comment, we have added the lack of information regarding the microbiological mechanisms as a limitation of the study to the Discussion section of the revised manuscript, as follows:

Line 391: “Second, the accurate taxonomy of the specific members of the *Porphyromonadaceae* that are involved should be characterized and they should be isolated and cultured, such that their functional roles in the host can be determined.”

Line 404: “Lastly and importantly, the use of FMT instead of ABX for patients with diarrhea should be pursued with caution, given that the present data were obtained from a study of

cattle. Its clinical application should be restricted to cattle until more comprehensive data are obtained in other species, including humans.”

367 and donors and recipients should be matched on the basis of their microbial profiles, which would permit customized FMT or defined microbial manipulations.

Again these are not humans and even in humans this kind of “matching” is still just a dream (and possibly even a wrong direction of thinking). If a gut microbiome is severely messed up (diarrhea) you simply need to reconstitute a working fermentative trophic network into the gut (preferably replacing whatever crap was in there previously). In the case of patients where there is just a slight degree of dysbiosis (but not a complete diarrheic mess) I suppose there might be more of a chance of that certain donors will not be successful in getting incorporated into the gut microbiome of the recipient (and that some others will have more success). In any case, when dealing with diarrheic calves, you’re just dealing with a mess (Figure 1 certainly gives that impression) and you simply need healthy calf donors (as you had here).

You might nonetheless in a larger study however certainly want to do more research which kinds of donors have the most success (if any big differences can be found). Perhaps some donors are indeed simply better overall than other donors (like in humans).

Response: In response to your comment, we have described a further direction of the FMT study in the Discussion section of the revised manuscript, as follows:

Line 393: “Third, the effectiveness of FMT should be validated in larger numbers of animals that exhibit moderate-to-severe diarrheic symptoms, and the host baseline physiology, characteristics, and gut microbiota should be characterized in more detail. In particular, the

specific bacterial species associated with a positive or negative response should be identified and isolated, which would permit customized FMT or defined microbial manipulations in the future.”

This study has also shown that FMT is easier to perform in livestock than human patients, and suggests that it is associated with improvements in both animal welfare and profitability.

Indeed, so I would do away with too much caution. Not only for these reasons, but also for reasons of using less antibiotics in livestock (in regards to antibiotic resistance development; very important in regards to human health) but also for learning FMT lessons that might also be applied in humans.

Response: We believe that we have fully responded to this point above and in the Introduction and Discussion sections, as follows:

Line 44: “Antibiotics have been widely used to treat or prevent diarrhea and promote growth in livestock⁴⁻⁶. Recently however, accumulating evidence has shown that the use of antibiotics in animal husbandry is associated with many adverse effects. The emergence of antibiotic-resistant bacteria and antibiotic residues in meat are recognized as major problems⁷⁻⁹. Due to their potential to kill both pathogenic and beneficial microbes, the use of wide-spectrum antibiotics can promote the colonization of the gut by pathogenic microbes, which can cause disease¹⁰⁻¹². Thus, antibiotic therapy of diarrheic calves often leads to the recurrence of serious diarrhea within days of starting it. The presence of an imbalance in the intestinal microbial community, followed by the administration of antibiotics, is capable of activating immune responses, inflammation, and peristalsis in the host gut, which causes diarrhea^{13, 14}. Importantly, a combination of recurrent diarrhea and antibiotic abuse in pre-

weaning calves may result in immaturity of the ruminal and intestinal microbiota, which has permanent negative effects on the digestion and absorption of dietary components during the fattening period¹⁵⁻¹⁷. Given that antibiotics also have detrimental impacts on ecology and food safety, their use should be reduced in farming and alternatives to antibiotic treatment identified.”

Line 404: “Lastly and importantly, the use of FMT instead of ABX for patients with diarrhea should be pursued with caution, given that the present data were obtained in a study of cattle. Its clinical application should be restricted to cattle until more comprehensive data are obtained in other species, including humans”.

10. Sullivan, A., Edlund, C. & Nord, C.E. Effect of antimicrobial agents on the ecological balance of human microflora. *Lancet Infect Dis* **1**, 101-114 (2001).
11. Buffie, C.G. & Pamer, E.G. Microbiota-mediated colonization resistance against intestinal pathogens. *Nat Rev Immunol* **13**, 790-801 (2013).
12. Ubeda, C. & Pamer, E.G. Antibiotics, microbiota, and immune defense. *Trends Immunol* **33**, 459-466 (2012).
13. Spees, A.M., Lopez, C.A., Kingsbury, D.D., Winter, S.E. & Baumler, A.J. Colonization Resistance: Battle of the Bugs or Menage a Trois with the Host? *Plos Pathog* **9** (2013).
14. Rivera-Chavez, F. et al. Depletion of Butyrate-Producing Clostridia from the Gut Microbiota Drives an Aerobic Luminal Expansion of Salmonella. *Cell Host Microbe* **19**, 443-454 (2016).

15. Ji, S. et al. Ecological Restoration of Antibiotic-Disturbed Gastrointestinal Microbiota in Foregut and Hindgut of Cows. *Front Cell Infect Microbiol* **8**, 79 (2018).
16. Wolin, M.J. Fermentation in the rumen and human large intestine. *Science* **213**, 1463-1468 (1981).
17. Russell, J.B. & Rychlik, J.L. Factors that alter rumen microbial ecology. *Science* **292**, 1119-1122 (2001).

544 Multiple variable correlation analysis Correlation analysis was used to identify the strength of the relationships between the relative abundance of the family Enterobacteriaceae or the family Porphyromonadaceae and the incidence of diarrhea in intermittently diarrheic calves. Pearson correlation coefficients and statistical significance (two-tailed) for each pair of variables were calculated using data from individual calves or merged data. Spearman's correlation analysis was used to identify associations with the bacterial taxa, with no specified statistical threshold.

With spearman rho correlation analyses I'm actually typically more interested in the correlation coefficients than in the p-values (as the p-value is dependent on the number of samples while the correlation coefficient is not so much affected). It's a good thing that in Figure 3 in the mini-heatmaps that the correlation coefficients are indeed given instead of p-values.

Response: Thank you for your valuable comments. *P*-values are presented in parentheses below the correlation coefficients for the mini-heatmap (Fig. 3f).

“Fig. 3 Discriminating microbial taxa associated with the incidence of diarrhea, and correlation analysis of gut *Enterobacteriaceae* or *Porphyromonadaceae* with the incidence of diarrhea

f The relative abundances of the families *Enterobacteriaceae* (left y-axis, orange) and *Porphyromonadaceae* (left y-axis, blue), and the BSS (right y-axis, brown) in each group over time (left panel). The Pearson correlation coefficients for multiple variants and statistical significance (two-tailed) for the relationships among BSS and the relative abundances of the families *Enterobacteriaceae* and *Porphyromonadaceae* were calculated (right panel). CON, control; ABX, antibiotic; FMT, fecal microbiota transplantation; BSS, Bristol stool scale; EN, *Enterobacteriaceae*; PO, *Porphyromonadaceae*.”

In regards to the bioinformatics pipeline, I would recommend perhaps switching to Dada2 or perhaps Mothur-Oligotyping-ARB, looking at ASVs or Oligotypes and then giving representative sequences of these ASVs or Oligotypes (so people can check the species themselves with their own taxonomic identification software if they are interested). This might provide a bit more detail than the current QIIME pipeline can provide. Also, QIIME2 now exists (so at least (eventually) switch to that). However, for the present study the current bioinformatics pipeline in general provided more than enough resolution as the results were so abundantly clear. Figure 5F could however have been more informative with more resolution.

Response: As you recommended, the bioinformatics pipeline was changed, and the Illumina MiSeq sequencing datasets were re-analyzed using the QIIME2 software package (version 2020.06). The imported paired reads were quality filtered, denoised, and merged using the plugin DADA2 to generate the ASV feature table. Taxonomic classification was performed using the plugin q2-feature-classifier, a taxonomic classifier plugin for the QIIME 2 microbiome analysis platform (<https://qiime2.org/>), which makes similar calculations using a scikit-learn naive Bayes classifier. Finally, taxonomy was assigned to the filtered ASVs using a pre-trained QIIME2-compatible SILVA version 132 database with 99% identity for the bacteria and representative sequences. All the data shown in Figs. 2, 3, and 5 are the results of new analyses. SourceTracker and PICRUSt analysis were also performed, and detailed information related to the bioinformatics analysis is provided under “*Fecal microbial profiling*” in the Methods section.

Line 567: “*Sequencing data analysis*”

The adapter sequences were trimmed from the raw fastq files, and the trimmed reads were demultiplexed according to the samples using the bcl2fastq2 conversion software version 2.20.0. (Illumina). The sorted reads were imported and processed using QIIME2 version 2020.06⁶⁴ for further bioinformatics analyses. The imported paired reads were quality filtered, denoised, and merged using the plugin DADA2⁶⁵ to generate the ASV feature table. Chimeric sequences and singleton ASVs were excluded from further analyses. Taxonomic classification was performed using the plugin q2-feature-classifier⁶⁶, a taxonomic classifier plugin for the QIIME 2 microbiome analysis platform (<https://qiime2.org/>), which makes similar calculations using a scikit-learn⁶⁷ naive Bayes classifier. Finally, taxonomy was assigned to filtered ASVs using a pre-trained QIIME2-compatible SILVA version 132 database⁶⁸, with 99% identity for the bacteria and representative sequences. To determine the species diversity in each sample, alpha and beta diversity analyses were performed using the plugin q2-diversity in QIIME2 version 2020.06, at a sampling depth of 28,147 for the preliminary trial set, 20,125 for the validation trial set, and 20,293 for the 12-month-old cattle set. Overviews of the Illumina-Miseq-generated datasets are provided in the supplementary tables as follows: for the preliminary study (54 samples), Supplementary Table 6; for the validation study (450 samples), Supplementary Tables 7 and 8. All 16S rRNA gene sequencing data have been deposited in the European Nucleotide Archive and are available under the accession number PRJEB35993. To compare bacterial communities between individuals and between groups, unweighted and weighted UniFrac outputs were assessed and visualized using unsupervised PCoA and distance histograms. Differences between groups were identified using PERMANOVA. Treatment-dependent features were identified using LefSe⁶⁹. A size-effect threshold of 3.5 on the logarithmic LDA score was used to identify discriminating taxa. The Mann–Whitney *U*-test was used to identify taxa that were

significantly different in their relative abundance among the groups. Details related to the analysis are described through Code availability.”

64. Bolyen, E. et al. Reproducible, interactive, scalable and extensible microbiome data science using QIIME 2 (vol 37, pg 852, 2019). *Nat Biotechnol* **37**, 1091-1091 (2019).
65. Callahan, B.J. et al. DADA2: High-resolution sample inference from Illumina amplicon data. *Nat Methods* **13**, 581-583 (2016).
66. Bokulich, N.A. et al. Optimizing taxonomic classification of marker-gene amplicon sequences with QIIME 2 's q2-feature-classifier plugin. *Microbiome* **6** (2018).
67. Pedregosa, F. et al. Scikit-learn: Machine Learning in Python. *J Mach Learn Res* **12**, 2825-2830 (2011).
68. Quast, C. et al. The SILVA ribosomal RNA gene database project: improved data processing and web-based tools. *Nucleic Acids Res* **41**, D590-D596 (2013).
69. Segata, N. et al. Metagenomic biomarker discovery and explanation. *Genome Biol* **12**, R60 (2011).

“Multiple variable correlation analysis

Correlation analysis was used to identify the strengths of the relationships between the relative abundance of the family *Enterobacteriaceae* or the family *Porphyromonadaceae* and the incidence of diarrhea in intermittently diarrheic calves. Pearson correlation coefficients and statistical significance (two-tailed) for each pair of variables were calculated using data from individual calves or merged data. Spearman’s correlation analysis was used to identify associations with the bacterial taxa, with no specified statistical threshold being set.

SourceTracker analysis

The tool SourceTracker was used to estimate the proportions of microbes from each source in a target microbial community (sink)⁷⁰. In the preliminary trial analysis, all the recipient samples were entered as individual sinks, while the donor feces, food pellets, and maternal milk were entered as individual sources. In the validation trial analysis, all the recipient samples were entered as individual sinks, while all the donor feces were entered as individual sources. The ASV feature tables and mapping files corresponding to each analysis set were used as input files, and the SourceTracker analysis was performed using default settings. Details related to the analysis are described through Code availability.”

70. Knights, D. et al. Bayesian community-wide culture-independent microbial source tracking. *Nat Methods* **8**, 761-U107 (2011).

“*PICRUSt analysis*

PICRUSt (<http://picrust.github.io>)³⁵, for predicting gene families based on the 16S rRNA gene composition, was used to characterize the functional profiles of the 12-month-old cattle gut microbial community. The constructed ASV feature table was converted into the PICRUSt format and normalized to 16S rRNA gene copy number to correct for over- and under-estimation of microbial abundance. The normalized dataset was analyzed using the KO (Kyoto Encyclopedia of Genes and Genomes orthology) dataset⁷¹.”

71. Kanehisa, M. & Goto, S. KEGG: Kyoto Encyclopedia of Genes and Genomes. *Nucleic Acids Res* **28**, 27-30 (2000).

Best regards,

Marcus de Goffau

REVIEWER COMMENTS

Reviewer #1 (Remarks to the Author):

The Authors have done an excellent job of revising their paper. They have included more analysis and responded well to all questions. It remains slightly unclear how the calves were housed before and after separation from their mother. Were there contact between the experimental calves (both inoculated and control) that could lead to cross contamination (within and between groups) ? I realize it is most of all a theoretical question whether each calf can be seen as the experimental unit. I accept the view that they can be seen as independent units even if they may have physical contact, as the major source for colonization of the gut is from the surroundings rather than other calves.

Collectively I find that the authors have done an impressive amount of work in both the initial submission and during the revision phase.

Reviewer #2 (Remarks to the Author):

I still like the paper and I'm looking forward to using Figure 1 during presentations about FMT once it is published (pictures tell much more in much less words and might also have a nice shock effect). I mainly have some points/clarifications in regards to the disturbance of fermentation efficiency during dysbiosis.

General comments:

You might want to explain to the readers somewhere what the reason is why source tracker still finds quite a lot of donor material pre-FMT (day 0).

In regards to the Porphyromonadaceae, it would be good to also publish the ASVs belonging to this family (and perhaps blast them to see what the closest match could be). There is currently some discussion in the conclusion but having the actual sequences, as produced by DADA2, would be valuable. I don't think that the "good" Porphyromonadaceae species in these calves are truly very similar to "bad" Porphyromonadaceae found in various studies with humans (same family absolutely does not mean same function). In general, an excel file with all ASVs (the good, the bad, the ugly and the neutral), and with all the abundances of all ASVs in all samples individually would be valuable.

When the gut microbiota is healthy (eubiotic status, line 371), a nutritional network is maintained that is suitable for fermentation of substances present in the intestine, but when the gut microbial balance is disrupted (dysbiotic status), a nutritional network with low fermentation efficiency is maintained.

1: A better term is probably “trophic network”. Secondly, it is not that a “trophic network” with low fermentation efficiency is “maintained” during dysbiosis; the problem is that a highly fermentatively efficient trophic network is disrupted during dysbiosis. What is left is an incoherent mess that is not very efficient anymore. This incoherent mess is what you visualize in Figure 1. During FMT the mess becomes coherent/structured again (restoration of cross-feeding between species & syntrophic interactions etc) whilst it remains a mess in the control and ABX group.

Similarly, the results of pediatric studies have suggested that a significant fecal loss of amino acids occurs in infant diarrhea: fecal free amino acid concentrations have been shown to be 10-fold higher during diarrhea than during periods of remission or in normal individuals.

The term “loss” in the 1st sentence might be a bit confusing (the 2nd part is crystal clear). Why not simply say: Similarly, the results of pediatric studies have shown that fecal free amino acid concentrations were 10-fold higher during diarrhea than during periods of remission or in normal individuals (references 52, 53).

In light of these previous findings, it is likely that an imbalance in the intestinal amino acids, which is followed by microbial dysbiosis, is associated with a higher risk of diarrhea in calves.

I think the chicken and the egg might have been mixed up here. You get an imbalance in the intestinal amino acids due to dysbiosis. This amino acid imbalance might then in turn however indeed have various negative effects.

Minor comments:

LEfSe was used to identify the bacterial families that were enriched in each group of cattle, and the Bacteroidaceae and 280 Porphyromonadaceae were found to discriminate the CON cattle; the Lachnospiraceae was found to discriminate the ABX cattle; the Christensenellaceae, Clostridiaceae, Peptostreptococcaceae, Dehalobacteriaceae and Coriobacteriaceae were found to 283 discriminate the FMT cattle (Fig. 5f).

“Was found to discriminate”: Is a bit vague. Discriminate from what? “More abundant” is probably simpler and clearer.

Best regards,

Marcus de Goffau

Responses to Reviewer's comments

(Longitudinal evaluation of fecal microbiota transplantation for ameliorating calf diarrhea
and improving growth performance)

(The authors' replies are written in blue)

Manuscript Number: NCOMMS-20-21979A

Comments from reviewer(s):

Referee #1 (Remarks to the Author):

The Authors have done an excellent job of revising their paper. They have included more analysis and responded well to all questions. It remains slightly unclear how the calves were housed before and after separation from their mother. Were there contact between the experimental calves (both inoculated and control) that could lead to cross contamination (within and between groups) ? I realize it is most of all a theoretical question whether each calf can be seen as the experimental unit. I accept the view that they can be seen as independent units even if they may have physical contact, as the major source for colonization of the gut is from the surroundings rather than other calves. Collectively I find that the authors have done an impressive amount of work in both the initial submission and during the revision phase.

Response: Thank you for your valuable comments. As described in the Methods, before weaning (60 days after birth) each mother-calf pair mostly lived separately, but physical contact was possible. After weaning (60 days after birth), all the calves were randomly mixed and housed in stalls containing five calves each. Although this carried a risk of cross contamination, it was considered more important to provide similar environments for each calf to minimize the effects of the stall. Nevertheless, it is worth noting that FMT-induced alterations in the gut microbiota were present in 12-month-old calves, as in 48-day-old calves. As per your comment, we have added information regarding the housing of the calves after their separation from their mothers to the Methods section of the revised manuscript, as follows:

Line 439: “*Animals*

All the calves had free access to food and water, and the mothers nurtured their calves in individual barns. Prior to weaning (60 days after birth), each calf was housed with their mother, and thereafter each mother-calf pair was separated. Although not all the calves were kept in the same space, a physical space was created that could be accessed by the calves, where they could eat their feed and interact with other calves. After weaning (60 days after birth), all the calves were randomly mixed and housed in stalls containing five calves each. All the groups were housed in stalls and fed the same appropriate diet for each stage of growth (Supplementary Table 3) to minimize any stall- or diet-induced inter-individual variation in their intestinal microbiota. All the barns were divided into 3 m by 3 m spaces. The floor was kept dry, and the individual buckets and feed bins were cleaned daily throughout the study.”

Referee #2 (Remarks to the Author):

I still like the paper and I'm looking forward to using Figure 1 during presentations about FMT once it is published (pictures tell much more in much less words and might also have a nice shock effect). I mainly have some points/clarifications in regards to the disturbance of fermentation efficiency during dysbiosis.

Response: Thank you for your critical review and valuable comments. As requested, we have revised the manuscript and included the data (ASVs belongs to the family *Porphyromonadaceae*) in the revised version of the manuscript. Our responses are set out below.

General comments:

You might want to explain to the readers somewhere what the reason is why source tracker still finds quite a lot of donor material pre-FMT (day 0).

Response: Similar results are apparent in the number of ASVs that were shared between the feces of healthy donor calves and those of the day 0 calves (Supplementary Fig. 3a,b).

Supplementary Fig. 3 Contributions of donor feces to the bacterial communities in the CON, ABX, and FMT calves

a–b The number of ASVs shared between the feces of healthy donor calves and the CON, ABX, and FMT calves on **(a)** day 0 and **(b)** day 48 after the start of treatment. The percentages of ASVs shared are shown.

We could not provide direct evidence during the study, but we believe that the ASVs (including donor material) identified in the intestines of day 0 calves comprise the core microbiota that are inherited from the mother. We believe that this core microbiota is commonly observed in the development of newborn calves. As per your comment, we have added the related content to the revised manuscript, as follows:

Line 161: “The results show that the mean contributions of the donor feces to the gut bacterial communities of CON, ABX, and FMT calves were similar on day 0 (Fig. 2l and Supplementary Fig. 3c). Thus, these intestinal ASVs represent the core microbiota that is inherited from the mother, and this core microbiota has frequently been identified during the development of newborn calves.”

In regards to the Porphyromonadaceae, it would be good to also publish the ASVs belonging to this family (and perhaps blast them to see what the closest match could be). There is currently some discussion in the conclusion but having the actual sequences, as produced by DADA2, would be valuable. I don't think that the “good” Porphyromonadaceae species in these calves are truly very similar to “bad” Porphyromonadaceae found in various studies with humans (same family absolutely does not mean same function). In general, an excel file with all ASVs (the good, the bad, the ugly and the neutral), and with all the abundances of all ASVs in all samples individually would be valuable.

Response: Thank you for your valuable comments. In response, we have added the data, including the ASVs that belong to the family *Porphyromonadaceae*, to the Source Data (Figure 3eiii).

When the gut microbiota is healthy (eubiotic status, line 371), a nutritional network is maintained that is suitable for fermentation of substances present in the intestine, but when the gut microbial balance is disrupted (dysbiotic status), a nutritional network with low fermentation efficiency is maintained.

1: A better term is probably “trophic network”. Secondly, it is not that a “trophic network” with low fermentation efficiency is “maintained” during dysbiosis; the problem is that a highly fermentatively efficient trophic network is disrupted during dysbiosis. What is left is an incoherent mess that is not very efficient anymore. This incoherent mess is what you visualize in Figure 1. During FMT the mess becomes coherent/structured again (restoration of cross-feeding between species & syntrophic interactions etc) whilst it remains a mess in the control and ABX group.

Response: In response to your comment, we have modified the sentence to read:

Line 368: “When the gut microbiota is healthy (eubiotic status), a trophic network is maintained that is suitable for the fermentation of substances present in the intestine, but when the gut microbial balance is disrupted (dysbiotic status), this trophic network is disrupted and the community shows poor fermentation efficiency⁵¹.”

Similarly, the results of pediatric studies have suggested that a significant fecal loss of amino acids occurs in infant diarrhea: fecal free amino acid concentrations have been shown to be 10-fold higher during diarrhea than during periods of remission or in normal individuals.

The term “loss” in the 1st sentence might be a bit confusing (the 2nd part is crystal clear). Why not simply say: Similarly, the results of pediatric studies have shown that fecal free amino acid concentrations were 10-fold higher during diarrhea than during periods of remission or in normal individuals (references 52, 53).

Response: We have modified the sentence to read:

Line 373: “Similarly, the results of pediatric studies have shown that fecal free amino acid concentrations are 10-fold higher during diarrhea than those during periods of remission or in normal individuals^{52, 53.}”

In light of these previous findings, it is likely that an imbalance in the intestinal amino acids, which is followed by microbial dysbiosis, is associated with a higher risk of diarrhea in calves.

I think the chicken and the egg might have been mixed up here. You get an imbalance in the intestinal amino acids due to dysbiosis. This amino acid imbalance might then in turn however indeed have various negative effects.

Response: Thank you for your valuable comment. We apologize for not expressing this concept appropriately. Therefore, we have modified the sentence, as follows:

Line 386: “In light of these previous findings, it is likely that microbial dysbiosis, which leads to an imbalance in the intestinal amino acids, is associated with a higher risk of diarrhea in calves.”

Minor comments:

LEfSe was used to identify the bacterial families that were enriched in each group of cattle, and the Bacteroidaceae and Porphyromonadaceae were found to discriminate the CON cattle; the Lachnospiraceae was found to discriminate the ABX cattle; the Christensenellaceae, Clostridiaceae, Peptostreptococcaceae, Dehalobacteriaceae and Coriobacteriaceae were found to discriminate the FMT cattle (Fig. 5f).

“Was found to discriminate”: Is a bit vague. Discriminate from what? “More abundant” is probably simpler and clearer.

Response: We have modified the sentence to read:

Line 278: “LEfSe was used to identify the bacterial families that were enriched in each group of cattle. This showed that the *Bacteroidaceae* and Porphyromonadaceae were more abundant in the CON cattle, the *Lachnospiraceae* were more abundant in the ABX cattle, and the *Christensenellaceae*, *Clostridiaceae*, *Peptostreptococcaceae*, *Dehalobacteriaceae*, and *Coriobacteriaceae* were more abundant in the FMT cattle (Fig. 5f).”

Best regards,

Marcus de Goffau